# High Probability Bound for Cross-Learning Contextual Bandits with Unknown Context Distributions

**Ruiyuan Huang** [1]   **Zengfeng Huang** [1] [2]

## Abstract

Motivated by applications in online bidding and sleeping bandits, we examine the problem of contextual bandits with cross learning, where the learner observes the loss associated with the action across all possible contexts, not just the current round's context. Our focus is on a setting where losses are chosen adversarially, and contexts are sampled i.i.d. from a specific distribution. This problem was first studied by Balseiro et al. (2019), who proposed an algorithm that achieves near-optimal regret under the assumption that the context distribution is known in advance. However, this assumption is often unrealistic. To address this issue, Schneider & Zimmert (2023) recently proposed a new algorithm that achieves nearly optimal expected regret. It is well-known that expected regret can be significantly weaker than high-probability bounds. In this paper, we present a novel, in-depth analysis of their algorithm and demonstrate that it actually achieves near-optimal regret with *high probability*. There are steps in the original analysis by Schneider & Zimmert (2023) that lead only to an expected bound by nature. In our analysis, we introduce several new insights. Specifically, we make extensive use of the weak dependency structure between different epochs, which was overlooked in previous analyses. Additionally, standard martingale inequalities are not directly applicable, so we refine martingale inequalities to complete our analysis.

---

[1]School of Data Science, Fudan University, Shanghai, China [2]Shanghai Innovation Institute. Correspondence to: Zengfeng Huang <huangzf@fudan.edu.cn>.

*Proceedings of the 42nd International Conference on Machine Learning*, Vancouver, Canada. PMLR 267, 2025. Copyright 2025 by the author(s).

## 1. Introduction

In the contextual bandits problem, a learner repeatedly observes a context, chooses an action, and incurs a loss specific to that action. The goal of the learner is to minimize the cumulative loss over the time horizon. The contextual bandits problem is a fundamental problem in online learning having broad applications in fields like online advertising, personalized recommendations, and clinical trials (Li et al., 2010; Kale et al., 2010; Villar et al., 2015).

We consider the cross-learning contextual bandits problem. In this setting, the learner not only observes the loss for the current action under the current context, but also observes the loss for the current action under all other contexts. This problem models many interesting scenarios. One such example is the problem of learning to bid in first-price auctions. In this problem the context is the bidder's private value for the item, while the action is the bid. The cross-learning structure comes from the fact that the bidder can deduce the utility of the bid under all contexts (i.e., the utility of the bid under different private valuations for the item). Other examples include multi-armed bandits with exogenous costs, dynamic pricing with variable costs, and learning to play in Bayesian games (Balseiro et al., 2019).

Technically, the most interesting setting for the cross-learning contextual bandits problem is when the losses are chosen adversarially but the contexts are i.i.d. samples from an *unknown* distribution $\nu$. Recently, Schneider & Zimmert (2023) gave an algorithm achieving nearly optimal $\widetilde{O}(\sqrt{KT})$ expected regret in this scenario.

Schneider & Zimmert (2023) designed a sophisticated algorithm that operates over multiple epochs to achieve near-optimal regret. A key technique in their analysis is to sidestep high-probability bounds and instead focus on bounding the expected summation to improve their results. As a consequence, their analysis only provides a bound that holds in expectation. It is not immediately clear whether this is due to limitations in the analysis or if the algorithm is inherently suboptimal. In any case, if we aim for a high-probability bound, fundamentally new insights are required.

In this paper, we show that the algorithm indeed achieves nearly optimal $\widetilde{O}(\sqrt{KT})$ regret with high-probability. The

key contribution of our paper is the following theorem.

**Theorem 1** (Informal). *The algorithm in Schneider & Zimmert (2023) yields a regret bound of order $\widetilde{O}(\sqrt{KT})$ with high probability for any policy $\pi$.*

In this section we only give the informal version of Theorem 1. The formal version can be found in Section 4.

### 1.1. Technical Overview

Our theorem is built on a new and more in-depth analysis of the algorithm in Schneider & Zimmert (2023). This new analysis introduces several new insights. In particular, we exploit the weak dependency structure between different epochs, which was overlooked in previous work. One difficulty of doing so is that standard martingale inequalities are not directly applicable, so we refine martingale inequalities to complete our analysis.

To prepare the readers for our new analysis, we first briefly introduce the algorithm in Schneider & Zimmert (2023). The algorithm in Schneider & Zimmert (2023) is an EXP3-type algorithm. The key novelty in their algorithm is the construction of the loss estimates $\widehat{\ell}$ used in the FTRL subroutine. Due to some technical problems we detail later, the algorithm decomposes the time horizon into epochs of equal length. In each epoch $e$, the algorithm first estimates the probability[1] $f_e(a)$ of observing the reward of each arm $a$ in epoch $e$ by an estimator $\widehat{f}_e(a)$, which is constructed exclusively from samples in epoch $e - 1$. Note that thanks to the cross-learning structure, the probability of observing the reward of each arm $a$ is independent of the contexts. The algorithm then constructs the loss estimates as an importance-weighted estimator with $\frac{1}{\widehat{f}_e(a)}$ as the importance weight.

Schneider & Zimmert (2023) showed that the performance of the algorithm depends on how well the empirical importance weight $\frac{1}{\widehat{f}_e(a)}$ concentrates around the expected importance weight $\frac{1}{f_e(a)}$. Since the estimator $\widehat{f}_e(a)$ is constructed exclusively from samples in a single epoch rather than the entire time horizon, the concentration $|\frac{1}{f_e(a)} - \frac{1}{\widehat{f}_e(a)}|$ is not tight enough. To achieve the desired $\widetilde{O}(\sqrt{KT})$ regret under a not tight enough concentration, Schneider & Zimmert (2023) bounds only the expected bias of importance estimator $\mathbb{E}[\frac{1}{f_e(a)} - \frac{1}{\widehat{f}_e(a)}]$ rather than providing a high-probability bias bound. Bounding only the expected bias gives a small enough bound, however, they can achieve a bound only on the expected regret from a bound on the expected bias.

We overcome this difficulty and show that their algorithm actually achieves a high-probability bound. Our key observation is that different epochs in their algorithm are only weakly dependent[2] on each other. Thus, the bias $\frac{1}{\widehat{f}_e(a)} - \frac{1}{f_e(a)}$ for each epoch $e$ is also only weakly dependent on each other. Therefore, although we cannot establish a small enough bound for the bias of a single epoch $\frac{1}{f_e(a)} - \frac{1}{\widehat{f}_e(a)}$, we can give a small enough bound for the cumulative bias across all epochs $\sum_e \frac{1}{f_e(a)} - \frac{1}{\widehat{f}_e(a)}$. We then use the bound on the cumulative bias to bound the cumulative regret.

In addition to utilizing the weak dependency structure between different epochs, we also address two further technical difficulties to establish our result. The first difficulty is that the existing regret decomposition is too crude to yield a high-probability bound. Schneider & Zimmert (2023) establish an $\widetilde{O}(\sqrt{KT})$ expected regret by decomposing the regret into different parts and bounding each part separately. Although their decomposition gives an $\widetilde{O}(\sqrt{KT})$ expected regret bound, it is too crude to derive a tight high-probability regret bound, even after utilizing the weak dependency structure. We carefully rearrange the regret decomposition to address this difficulty.

Secondly, we cannot simply apply standard martingale concentration inequalities to $\sum_e \frac{1}{f_e(a)} - \frac{1}{\widehat{f}_e(a)}$ to bound its deviation. The main problem is that the random variable $\frac{1}{f_e(a)} - \frac{1}{\widehat{f}_e(a)}$ is not almost surely bounded by a constant, which makes standard martingale concentration inequalities inapplicable. We introduce a surrogate sequence of random variables as a bridge to address this problem. We bound the sum over the surrogate sequence, and show that the sum over the real sequence is equal to the surrogate sequence with high probability.

### 1.2. Related Works

The cross-learning contextual bandits problem was first proposed in Balseiro et al. (2019). They achieve the nearly optimal $\widetilde{O}(\sqrt{KT})$ regret under two scenarios: (1) when both losses and contexts are stochastic, and (2) when losses are adversarial and contexts are stochastic with a known distribution. When losses are adversarial and contexts are stochastic with an unknown distribution, they only achieve the suboptimal $\widetilde{O}(K^{1/3}T^{2/3})$ regret. More recently, Schneider & Zimmert (2023) gave a new algorithm that achieves the nearly optimal $\widetilde{O}(\sqrt{KT})$ regret in expectation under adversarial losses and stochastic contexts with an unknown

---

[1]For technical reasons, in the actual algorithm, the value $f_e(a)$ actually represents the probability of observing the reward of each arm $a$ in epoch $e + 2$. For ease of understanding, here we instead let it represent the probability of observing the reward of each arm $a$ in each epoch $e$.

[2]The term "weakly dependent" is an informal term commonly used in the context of concentration inequalities. It generally refers to a sequence of random variables that, while not fully independent, forms a martingale, allowing us to derive concentration inequalities akin to those for independent random variables.

distribution.

An important application of the cross-learning contextual bandits problem, which is also the primary motivation for proposing this problem in Balseiro et al. (2019), is to solve the problem of learning to bid in first-price auctions. In this problem the context is the bidder's private value for the item, while the action is the bid. The cross learning structure comes from the fact that the bidder can deduce the utility of the bid under all contexts (i.e., the utility of the bid under different private valuations for the item).

Balseiro et al. (2019) used the cross-learning contextual bandits problem to model the bidding problem and obtained an $O(T^{3/4})$ regret bound for bidders with an unknown value distribution participating in adversarial first-price auctions, where the only feedback is whether the bidder wins the auction. Later, many works studied different settings of the bidding in first price auctions problem. For example, Han et al. (2020b) considered the problem with censored feedback, where each bidder observes the winning bid. Han et al. (2020a) considered the scenario when the value is also adversarial. Ai et al. (2022); Wang et al. (2023) considered the problem under budget constraints. In all these scenarios, the cross learning structure between different values is an essential component of the analysis.

Another interesting application of the cross-learning contextual bandits problem is the sleeping bandits problem (Kleinberg et al., 2010; Neu & Valko, 2014; Kale et al., 2016; Saha et al., 2020). In this problem, a certain set of arms is unavailable in each round. The sleeping bandits problem is motivated by instances like some items might go out of stock in retail stores or on a certain day some websites could be down. When losses are adversarial and availabilities are stochastic, previous work either requires exponential computing time (Kleinberg et al., 2010; Neu & Valko, 2014) or results in suboptimal regret (Kale et al., 2016; Saha et al., 2020). The first computationally efficient algorithm with optimal regret $\widetilde{O}(\sqrt{KT})$ is proposed in Schneider & Zimmert (2023) by modeling the problem as a cross-learning contextual bandit.

We also note that handling unknown context distributions is a common and challenging problem across various contextual bandit problems. For example, in the adversarial linear contextual bandits problem (Neu & Olkhovskaya, 2020), the linear MDP problem (Dai et al., 2023), and the oracle-based adversarial contextual bandits problem (Syrgkanis et al., 2016), existing algorithms often rely on knowledge of the context distribution. Removing the reliance on knowledge of the context distribution is typically non-trivial (Liu et al., 2023; Dai et al., 2023).

Recently, Hanna et al. (2023) proposed a method for stochastic linear contextual bandits that maps a multi-context problem to a single-context problem. Unfortunately, their approach cannot be directly applied to our problem for two reasons. First, their method is designed for stochastic bandits, whereas we deal with adversarial bandits. Second, their approach is limited to linear contextual bandits. Whether it can be adapted, with certain modifications, to address our problem remains an intriguing question.

## 2. Problem Statament

We study a contextual $K$-armed bandit problem over $T$ rounds, with contexts belonging to the set $[C]$. At the beginning of the problem, an oblivious adversary selects a sequence of losses $\ell_{t,c}(a) \in [0,1]$ for every round $t \in [T]$, every context $c \in [C]$, and every arm $a \in [K]$. In each round $t$, we begin by sampling a context $c_t \sim \nu$ i.i.d. from an unknown distribution $\nu$ over $[C]$, and we reveal this context to the learner. Based on this context, the learner selects an arm $a_t \in [K]$ to play. The adversary then reveals the function $\ell_{t,c}(a_t)$, and the learner suffers loss $\ell_{t,c_t}(a_t)$. Notably, the learner observes the loss for every context $c \in [C]$, but only for the arm $a_t$ they actually played.

We aim to design learning algorithms that minimize regret. Fix a policy $\pi : [C] \to [K]$. With a slight abuse of notation, we also denote $\pi_c = e_k \in \Delta([K])$ for each $c \in [C]$. The (unexpected) regret with respect to policy $\pi$ is

$$\mathrm{Reg}(\pi) = \sum_{t=1}^{T} \ell_{t,c_t}(a_t) - \ell_{t,c_t}(\pi_{c_t}).$$

Schneider & Zimmert (2023) designed an algorithm that achieves an expected regret bound of $\mathbb{E}\left[\mathrm{Reg}(\pi)\right] \leq \widetilde{O}(\sqrt{KT})$ for any policy $\pi$. We will show that the algorithm in Schneider & Zimmert (2023) actually provides a high-probability regret bound.

## 3. The Algorithm in Schneider and Zimmert (2023)

In this section, we briefly recap the intuition behind the algorithm proposed in Schneider & Zimmert (2023) and redescribe the algorithm formally to prepare the readers for our new analysis.

### 3.1. Intuition behind Schneider and Zimmert (2023)

The algorithm proposed in Schneider & Zimmert (2023) is an EXP3-type algorithm. Similar to the well-known EXP3 algorithm, at each round $t$, the algorithm generates a distribution using an FTRL subroutine

$$p_{t,c} = \arg\min_{p \in \Delta([K])} \left\langle p, \sum_{s=1}^{t-1} \widehat{\ell}_{s,c} \right\rangle - \frac{1}{\eta} F(p)$$

for each context $c$, where $F(p) = \sum_{i=1}^{K} p_i \log(p_i)$ is the unnormalized negative entropy, $\eta$ is a learning rate, and $\widehat{\ell}$ are loss estimates to be defined later. The algorithm then essentially samples the action $a_t$ to be played in round $t$ from distribution $p_{t,c_t}$.

The key novelty in Schneider & Zimmert (2023) lies in the construction of the loss estimates $\widehat{\ell}$. An intuitive construction is defined as follows:

$$\widetilde{\ell}_{t,c}(a) = \frac{\ell_{t,c}(a)}{\mathbb{E}_{c \sim \nu}[p_{t,c}(a)]} \mathbb{1}(a_t = a).$$

That is, it uses the classic importance-weighted estimator with $\mathbb{E}_{c \sim \nu}[p_{t,c}(a)]$ as the importance[3]. A straightforward analysis shows that this estimator yields a regret bound of $\widetilde{O}(\sqrt{KT})$. However, the denominator term $\mathbb{E}_{c \sim \nu}[p_{t,c}(a)]$ is uncomputable because we do not know the distribution of contexts $\nu$. One may attempt to circumvent this issue by replacing the expected importance $\mathbb{E}_{c \sim \nu}[p_{t,c}(a)]$ with the empirical importance $\frac{1}{t} \sum_{s=1}^{t} p_{t,c_s}(a)$. It is not hard to see that whether we achieve the desired $\widetilde{O}(\sqrt{KT})$ regret depends on how well the empirical importance weight $\frac{1}{\frac{1}{t} \sum_{s=1}^{t} p_{t,c_s}(a)}$ concentrates around the expected importance weight $\frac{1}{\mathbb{E}_{c \sim \nu}[p_{t,c}(a)]}$. However, the empirical importance weight $\frac{1}{\frac{1}{t} \sum_{s=1}^{t} p_{t,c_s}(a)}$ may not concentrate well around the expected importance weight $\frac{1}{\mathbb{E}_{c \sim \nu}[p_{t,c}(a)]}$. This is because the probability vector $p_{t,c}$ is not independent of the previous contexts $c_s$, which makes standard concentration inequalities inapplicable.

To address this difficulty, Schneider & Zimmert (2023) divides the time horizon into epochs of equal length $L$. At the end of each epoch $e$, the algorithm stores the FTRL distribution at the current time $t = eL$ in a new distribution $s_e$; that is, it takes $s_{e,c}(a) = p_{t,c}(a)$ for each context $c$ and each arm $a$. The algorithm further decouples the distribution played by the algorithm and the distribution used to estimate the loss vector. For each time $t$ in epoch $e + 2$, the algorithm observes the loss $\ell_{t,c}(a)$ for each arm $a$ and context $c$ with probability $f_e(a) \triangleq \mathbb{E}_{c \sim \nu}[s_{e,c}(a)/2]$. The algorithm then estimates the expected importance $f_e(a)$ using an empirical importance $\widehat{f}_e(a)$ constructing solely from contexts in epoch $e + 1$. Finally, the algorithm constructs $\widehat{\ell}_{t,c}(a)$ as an importance-weighted estimator with $\widehat{f}_e(a)$ serving as the importance.

The advantage of their construction is that the empirical importance weight $\frac{1}{\widehat{f}_e(a)}$ concentrates around the expected importance weight $\frac{1}{f_e(a)}$ now. This concentration ensures that the loss estimates $\widehat{\ell}_{t,c}(a)$ are good estimates of the true

---

[3]In this paper we call terms like $\frac{1}{\mathbb{E}_{c \sim \nu}[p_{t,c}(a)]}$ as the *importance weight* and call terms like $\mathbb{E}_{c \sim \nu}[p_{t,c}(a)]$ as the *importance*.

losses $\ell_{t,c}(a)$. And this concentration is achieved because the algorithm constructs the estimator using only samples from epoch $e + 1$, which are independent of the estimand.

### 3.2. A Formal Description of the Algorithm in Schneider and Zimmert (2023)

In this subsection we describe the algorithm in Schneider & Zimmert (2023) formally for the sake of completeness. Readers familiar with Schneider & Zimmert (2023) can skip this subsection safely.

In each round $t$, the algorithm generates a distribution from an FTRL subroutine:

$$p_{t,c} = \arg\min_{p \in \Delta([K])} \left\langle p, \sum_{s=1}^{t-1} \widehat{\ell}_{s,c} \right\rangle - \frac{1}{\eta} F(p)$$

for each context $c$, where $F(p) = \sum_{i=1}^{K} p_i \log(p_i)$ is the unnormalized negative entropy, $\eta$ is the learning rate, and $\widehat{\ell}$ are loss estimates to be defined later. The algorithm will not sample the action $a_t$ played in round t directly from $p_t$ but from a distribution $q_t$ to be defined later.

To construct loss estimates $\widehat{\ell}$, the algorithm divides the time horizon into epochs of equal length $L$. We let $\mathcal{T}_e$ to denote the set of rounds in the $e$-th epoch. At the end of each epoch, the algorithm takes a single snapshot of the underlying FTRL distribution $p_t$ for each context and arm. That is, the algorithm takes

$$s_{e+2,c}(a) = p_{eL,c}(a)$$

and takes

$$s_{1,c}(a) = s_{2,c}(a) = \frac{1}{K}.$$

The index of the snapshot $s_{e+2}$ for epoch $e$ is $e + 2$ because we use the snapshot $s_{e+2}$ in epoch $e + 2$.

For each round $t \in \mathcal{T}_e$, the algorithm observes the loss function of arm $a$ with probability $f_e(a) = \mathbb{E}_{c \sim \nu}[s_{e,c}(a)/2]$. This is guaranteed by the following rejection sampling procedure: we first play an arm according to the distribution

$$q_{t,c_t} = \begin{cases} p_{t,c_t} & \text{if } \forall a \in [K] : p_{t,c_t}(a) \geq s_{e,c_t}(a)/2 \\ s_{e,c_t} & \text{otherwise.} \end{cases}$$

After playing arm $a$ according to $q_{t,c_t}$, the learner samples a Bernoulli random variable $S_t$ with probability $\frac{s_{e,c_t}(a)}{2q_{t,c_t}(a)}$. If $S_t = 0$, the learner ignores the feedback from this round; otherwise, they use this loss.

The only remaining unspecified part is how to construct the loss estimates. We group all timesteps into consecutive pairs of two. In each pair of consecutive timesteps, we sample from the same distribution and randomly use one to calculate a loss estimate and the other to estimate the sampling

frequency. To be precise, let $\mathcal{T}_e^f$ denote the timesteps selected for estimating the sampling frequency and $\mathcal{T}_e^\ell$ denote the timesteps used to estimate the losses. Then we define

$$\widehat{f}_e(a) = \frac{1}{\left|\mathcal{T}_{e-1}^f\right|} \sum_{t \in \mathcal{T}_{e-1}^f} \frac{s_{e,c_t}(a)}{2}$$

which is an unbiased estimator of $f_e(a)$. The loss estimators are defined as follows:

$$\widehat{\ell}_{t,c}(a) = \frac{2\ell_{t,c}(a)}{\widehat{f}_e(a) + \frac{3}{2}\gamma} \mathbb{1}\left(A_t = a \wedge S_t \wedge t \in \mathcal{T}_e^\ell\right)$$

where $\gamma$ is a confidence parameter to be specified later.

The algorithm is summarized in Algorithm 1. Furthermore, Schneider & Zimmert (2023) showed that the algorithm achieves an expected regret bound of $\widetilde{O}(\sqrt{KT})$.

---

**Algorithm 1** The algorithm for the cross-learning problem in Schneider & Zimmert (2023)

**Input:** Parameters $\eta, \gamma > 0$ and $L < T$.
$\widehat{f}_2 \leftarrow 0$
**for** $t = 1, \ldots, L$ **do**
 Observe $c_t$
 Play $A_t \sim s_{1,c_t}$
 $\widehat{f}_2 \leftarrow \widehat{f}_2 + \frac{s_{2,c_t}}{2L}$
**for** $e = 2, \ldots, T/L$ **do**
 $\widehat{f}_{e+1} \leftarrow 0$
 **for** $t = (e-1)L + 1, t = (e-1)L + 3, \ldots, eL - 1$ **do**
  $p_{t,c} \leftarrow \arg\min_{p \in \Delta([K])} \left(\left\langle p, \sum_{s=1}^{t-1} \widehat{\ell}_s(c)\right\rangle - \eta^{-1} F(p)\right)$
  **for** $t' = t, t+1$ **do**
   Observe $c_{t'}$
   **if** $p_{t,c_{t'}}(a) \geq s_{e,c_{t'}}(a)/2$ *for all* $a \in [K]$ **then**
    Set $q_{t',c_{t'}} = p_{t,c_{t'}}$
   **else**
    Set $q_{t',c_{t'}} = s_{e,c_{t'}}$
   Play $A_{t'} \sim q_{t',c_{t'}}$
   Observe $\ell_{t',A_{t'}}$
  $t_f, t_\ell \leftarrow \mathsf{RandPerm}(t, t+1)$
  $\widehat{f}_{e+1} \leftarrow \widehat{f}_{e+1} + \frac{s_{e+1,c_{t_f}}}{2(L/2)}$
  Sample $S_t \sim \mathcal{B}\left(\frac{s_{e,c_{t_\ell}}(A_{t_\ell})}{2q_{t,c_{t_\ell}}(A_{t_\ell})}\right)$
  Set $\widehat{\ell}_{t_\ell,c}(a) = \frac{2\ell_{t_\ell,c}(a)}{\widehat{f}_e(a) + \frac{3}{2}\gamma} \mathbb{1}(A_t = a, S_t = 1)$
 $s_{e+2} \leftarrow p_t$

---

## 4. Main Result and Analysis

The main result of our paper is the following theorem.

**Theorem 1** (Formal). *For any $\delta \in (0,1)$, Algorithm 1 with parameters choice $\iota = 2\log(8KT\frac{1}{\delta})$, $L = \sqrt{\frac{\iota KT}{\log(K)}} =$*

$\widetilde{\Theta}(\sqrt{KT \log \frac{1}{\delta}})$, $\gamma = \frac{16\iota}{L} = \widetilde{\Theta}(\sqrt{\frac{\log(1/\delta)}{KT}})$, *and* $\eta = \frac{\gamma}{2(2L\gamma + \iota)} = \widetilde{\Theta}(1/\sqrt{KT \log(1/\delta)})$ *yields a regret bound of*

$$\mathrm{Reg}(\pi) = \widetilde{O}\left(\sqrt{KT \log \frac{1}{\delta}}\right)$$

*with probability at least $1 - \delta$ for any policy $\pi$.*

In what follows, we briefly overview our proof of Theorem 1. The full proof can be found in the appendix.

### 4.1. Regret Decomposition

Denote the set of all timesteps used to estimate the frequency as $\mathcal{T}^f$ and denote the set of all timesteps used to estimate the losses as $\mathcal{T}^\ell$. For each $t \in \mathcal{T}_e$, we define $\widetilde{\ell}_{t,c}(a) = \frac{2\ell_{t,c}(a)}{f_e(a)+\gamma} \mathbb{1}\left(A_t = a \wedge S_t \wedge t \in \mathcal{T}_e^\ell\right)$. We decompose regret $\mathrm{Reg}(\pi) = \sum_{t=1}^T \ell_{t,c_t}(a_t) - \ell_{t,c_t}(\pi_{c_t})$ as follows:

$$\mathrm{Reg}(\pi)$$
$$= \underbrace{\sum_{t=1}^T \left(\ell_{t,c_t}(a_t) - \ell_{t,c_t}(\pi_{c_t})\right) - 2\sum_{t \in \mathcal{T}^\ell}\left(\ell_{t,c_t}(a_t) - \ell_{t,c_t}(\pi_{c_t})\right)}_{\mathbf{bias}_1}$$
$$+ \underbrace{2\sum_{t \in \mathcal{T}^\ell}\left(\ell_{t,c_t}(a_t) - \ell_{t,c_t}(\pi_{c_t}) - \sum_c \Pr(c)\langle p_{t,c} - \pi_c, \ell_{t,c}\rangle\right)}_{\mathbf{bias}_2}$$
$$+ \underbrace{2\sum_{t \in \mathcal{T}^\ell}\sum_c \Pr(c)\langle p_{t,c} - \pi_c, \widehat{\ell}_{t,c}\rangle}_{\mathbf{ftrl}}$$
$$+ \underbrace{2\sum_{t \in \mathcal{T}^\ell}\sum_c \Pr(c)\left\langle p_{t,c}, \ell_{t,c} - \widetilde{\ell}_{t,c}\right\rangle}_{\mathbf{bias}_3}$$
$$+ \underbrace{2\sum_{t \in \mathcal{T}^\ell}\sum_c \Pr(c)\left\langle p_{t,c}, \widetilde{\ell}_{t,c} - \widehat{\ell}_{t,c}\right\rangle}_{\mathbf{bias}_4}$$
$$+ \underbrace{2\sum_{t \in \mathcal{T}^\ell}\sum_c \Pr(c)\left\langle \pi_c, \widehat{\ell}_{t,c} - \ell_{t,c}\right\rangle}_{\mathbf{bias}_5}.$$

In our decomposition, the $\mathbf{bias}_1$ term refers to the bias introduced by replacing the regret over the entire time horizon with that over $\mathcal{T}^\ell$, and the $\mathbf{bias}_2$ term refers to the bias introduced by replacing regret with its linearization. Both of these terms are not hard to bound using standard concentration inequalities. Furthermore, the $\mathbf{ftrl}$ and $\mathbf{bias}_3$ terms are standard in the analysis of high-probability bounds for bandit algorithms. These two terms are not hard to bound using techniqes from EXP3-IX (Neu, 2015; Schneider &

Zimmert, 2023). The **bias**$_4$ and **bias**$_5$ terms correspond to the bias introduced by constructing the importance estimator $\widehat{f}_e(a)$. These two terms are the terms of interest to bound.

Our decomposition is different from the decomposition in Schneider & Zimmert (2023). This difference is essential for deriving a high-probability bound. The key difference lies in the **bias**$_5$ term here. This term saves a $\sum_{t\in\mathcal{T}^\ell}\sum_c \Pr(c)\left\langle \pi_c, \widetilde{\ell}_{t,c} - \ell_{t,c}\right\rangle$ term from the **bias**$_2$ term in the decomposition given by Schneider & Zimmert (2023), which is crucial for deriving a high-probability bound.

### 4.2. Identifying a Prototypical Term

The terms of interest to bound are **bias**$_4$ and **bias**$_5$. These two terms can be bounded using similar methods. Here we take the **bias**$_5$ term as a prototypical term and give a sketch of its analysis. Details can be found in the appendix.

To bound **bias**$_5$, we define a filtration $\{\mathcal{H}_t\}_t$ such that the $\sigma$-algebra $\mathcal{H}_t$ for each time step $t$ is generated by all randomness before time $t$. Next, we decompose **bias**$_5$ as

$$\sum_{t\in\mathcal{T}^\ell}\sum_c \Pr(c)\left\langle \pi_c, \widehat{\ell}_{t,c} - \ell_{t,c}\right\rangle$$

$$= \sum_{t\in\mathcal{T}^\ell}\sum_c \Pr(c)\left(\widehat{\ell}_{t,c}(\pi_c) - \ell_{t,c}(\pi_c)\right)$$

$$= \sum_{t\in\mathcal{T}^\ell}\sum_c \Pr(c)\left(\mathbb{E}\left[\widehat{\ell}_{t,c}(\pi_c)\,\middle|\,\mathcal{H}_{t-1}\right] - \ell_{t,c}(\pi_c)\right)$$

$$+ \sum_{t\in\mathcal{T}^\ell}\sum_c \Pr(c)\left(\widehat{\ell}_{t,c}(\pi_c) - \mathbb{E}\left[\widehat{\ell}_{t,c}(\pi_c)\,\middle|\,\mathcal{H}_{t-1}\right]\right).$$

In this decomposition, the two terms correspond to different components of the bias of the estimator $\widehat{\ell}_{t,c}$. The first term $\sum_{t\in\mathcal{T}^\ell}\sum_c \Pr(c)\left(\mathbb{E}\left[\widehat{\ell}_{t,c}(\pi_c)\,\middle|\,\mathcal{H}_{t-1}\right] - \ell_{t,c}(\pi_c)\right)$ corresponds to the bias introduced by constructing the importance estimator $\widehat{f}_e(a)$. The second term $\sum_{t\in\mathcal{T}^\ell}\sum_c \Pr(c)\left(\widehat{\ell}_{t,c}(\pi_c) - \mathbb{E}\left[\widehat{\ell}_{t,c}(\pi_c)\,\middle|\,\mathcal{H}_{t-1}\right]\right)$ corresponds to the bias introduced from the randomness in sampling $a_t$ from $q_t$. Once again, the analyses of these two terms follow the same principle. We take the first term

$$\sum_{t\in\mathcal{T}^\ell}\sum_c \Pr(c)\left(\mathbb{E}\left[\widehat{\ell}_{t,c}(\pi_c)\,\middle|\,\mathcal{H}_{t-1}\right] - \ell_{t,c}(\pi_c)\right)$$

as a prototypical term and give a sketch of its analysis for the sake of simplicity. Details can be found in the appendix.

### 4.3. Analysis of the Prototypical Term

To bound the term

$$\sum_{t\in\mathcal{T}^\ell}\sum_c \Pr(c)\left(\mathbb{E}\left[\widehat{\ell}_{t,c}(\pi_c)\,\middle|\,\mathcal{H}_{t-1}\right] - \ell_{t,c}(\pi_c)\right),$$

we will use the key observation mentioned at Section 1.1: different epochs in Algorithm 1 are only weakly dependent on each other. To use this observation rigorously, we introduce an important technical tool. With a slight abuse of notation, we define a filtration $\{\mathcal{H}_e\}_e$, in which for each epoch $e$, the $\sigma$-algebra $\mathcal{H}_e$ is generated by all randomness in epochs $1,\ldots,e-1$ and the randomness in $\mathcal{T}_e^\ell$. That is, the $\sigma$-algebra $\mathcal{H}_e$ is generated precisely by the context $c_t$, the random seed used in sampling $a_t \sim q_{t,c_t}$, and the random seed used in sampling $S_t \sim \mathcal{B}\left(\frac{s_{e,c_t}(a_t)}{2q_{t,c_t}(a_t)}\right)$ for $t \leq (e-1)L$ and $t \in \mathcal{T}_e^\ell$. Note that for each epoch $e$, the $\sigma$-algebra $\mathcal{H}_e$ excludes the randomness in $\mathcal{T}_e^{\mathrm{f}}$. This exclusion is crucial for characterizing the weak dependence structure between epochs.

Given this filtration, we consider the cumulative bias in each epoch. For each epoch $e$, we define a random variable

$$\mathrm{Bias5}_e \triangleq \sum_{t\in\mathcal{T}_e^\ell}\sum_c \Pr(c)\left(\mathbb{E}\left[\widehat{\ell}_{t,c}(\pi_c)\,\middle|\,\mathcal{H}_{t-1}\right] - \ell_{t,c}(\pi_c)\right).$$

Then the prototypical term $\sum_{t\in\mathcal{T}^\ell}\sum_c \Pr(c)\left(\mathbb{E}\left[\widehat{\ell}_{t,c}(\pi_c)\,\middle|\,\mathcal{H}_{t-1}\right] - \ell_{t,c}(\pi_c)\right)$ is exactly $\sum_e \mathrm{Bias5}_e$. Our key observation is that, not only

$$\mathbb{E}\left[\sum_{t\in\mathcal{T}^\ell}\sum_c \Pr(c)\left(\mathbb{E}\left[\widehat{\ell}_{t,c}(\pi_c)\,\middle|\,\mathcal{H}_{t-1}\right] - \ell_{t,c}(\pi_c)\right)\right] \leq 0$$

as shown in Schneider & Zimmert (2023), but also

$$\sum_e \mathbb{E}\left[\mathrm{Bias5}_e\,\middle|\,\mathcal{H}_e\right] \sim -\sum_e \frac{\gamma}{f_e(\pi_c)+\gamma}.$$

This key observation improves the inequality in Schneider & Zimmert (2023) in two ways. Firstly, our bound holds for conditional expectations across epochs, which opens the door to applying martingale concentration inequalities across epochs. Secondly, our new decomposition improves the upper bound from $0$ to $-\sum_e \frac{\gamma}{f_e(\pi_c)+\gamma}$. This improvement is essential for deriving a high probability bound.

Given the new bound $\sum_e \mathbb{E}\left[\mathrm{Bias5}_e\,\middle|\,\mathcal{H}_e\right] \sim -\sum_e \frac{\gamma}{f_e(\pi_c)+\gamma}$, we only need to bound the deviation $\sum_e \mathrm{Bias5}_e - \mathbb{E}\left[\mathrm{Bias5}_e\,\middle|\,\mathcal{H}_e\right]$ to get an upper bound on $\sum_e \mathrm{Bias5}_e$. However, we cannot directly apply standard martingale concentration inequalities to $\sum_e \mathrm{Bias5}_e - \mathbb{E}\left[\mathrm{Bias5}_e\,\middle|\,\mathcal{H}_e\right]$. This is because we need to assume that the random variable $|\mathrm{Bias5}_e| \leq 2L$ almost surely to get a tight enough concentration bound when applying standard martingale concentration inequalities. However, this is not the case. The random variable $\mathrm{Bias5}_e$ exceeds the constant $2L$ with a small but positive probability. This unboundness prevents us from getting a tight enough concentration bound when applying standard martingale concentration inequalities.

To overcome this problem, we consider the indicator function

$$F_e \triangleq \mathbb{1}\left(\forall a, \left|\widehat{f}_e(a) - f_e(a)\right| \leq 2\max\left\{\sqrt{\frac{f_e(a)\iota}{L}}, \frac{\iota}{L}\right\}\right)$$

defined in Schneider & Zimmert (2023). We show that we also have $\sum_e \mathbb{E}\left[\text{Bias5}_e\, F_e \,|\, \mathcal{H}_e\right] \sim -\sum_e \frac{\gamma}{f_e(\pi_c)+\gamma}$ and that the random variable $|\text{Bias5}_e\, F_e| \leq 2L$ almost surely. Thus, we can use standard martingale concentration inequalities to get a tight enough concentration bound on $\sum_e \text{Bias5}_e\, F_e - \mathbb{E}\left[\text{Bias5}_e\, F_e \,|\, \mathcal{H}_e\right]$ and further bound $\sum_e \text{Bias5}_e\, F_e$. Finally, we have that $\sum_e \text{Bias5}_e\, F_e = \sum_e \text{Bias5}_e$ with high probability. Thus, a high probability bound on $\sum_e \text{Bias5}_e\, F_e$ transfers to a high probability bound on $\sum_e \text{Bias5}_e$.

## 5. Experiments

We conduct a simple experiment to show the robustness and efficiency of Algorithm 1. We consider the following adversarial contextual bandit instance. In round $t \in [T]$, the loss $\ell_{t,c}(a)$ for arm $a \in [K]$ under context $c \in [C]$ is sampled from a Bernoulli distribution with expectation $|\cos(a+c)t \cdot \sin a|$ where $a$, $c$, and $t$ are treated as integer-valued inputs to the trigonometric function.

We compare Algorithm 1 against the standard **One-Bandit-per-Context (OBPC)** algorithm (Lattimore & Szepesvári, 2020) on this problem instance. The OBPC baseline algorithm is a standard algorithm for vanilla adversarial contextual bandit problems. This baseline algorithm maintains $C$ independent instances of the standard Exp3 bandit algorithm, one for each of the $C$ contexts. When a context $c_t$ is observed, the baseline algorithm consults the specific bandit instance associated with $c_t$ to choose an arm and subsequently updates only that instance. This approach does not share information across different contexts. Despite its simplicity, the OBPC baseline algorithm is already minimax-optimal on vanilla adversarial contextual bandit problems.

The parameters for our experiments were set as follows. We set the time horizon $T = 10^4$, the number of arms $K = 9$, the number of contexts $C = 1000$. The contexts are sampled uniformly at random from $[C]$ in each round, i.e., $c_t \sim \text{Uniform}([C])$. Our results are shown in Figure 1.

The results depicted in Figure 1 clearly show that, compared with the OBPC baseline algorithm, Algorithm 1 consistently achieves a significantly lower average cumulative loss and is also more robust.

## 6. Conclusions

We reanalyze the algorithm proposed by Schneider & Zimmert (2023) and show that it actually achieves near-optimal regret with *high probability* for the cross-learning contextual bandits problem when the losses are chosen adversarially but the contexts are i.i.d. sampled from an *unknown* distribution. Our key technique is utilizing the weak dependency structure between different epochs for an algorithm executing over multiple epochs. It is of interest to investigate that whether this techniques is applicable for deriving high probability bounds for algorithms executing over multiple epochs in other problems.

## Impact Statement

This paper presents work whose goal is to advance the field of Machine Learning. There are many potential societal consequences of our work, none which we feel must be specifically highlighted here.

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

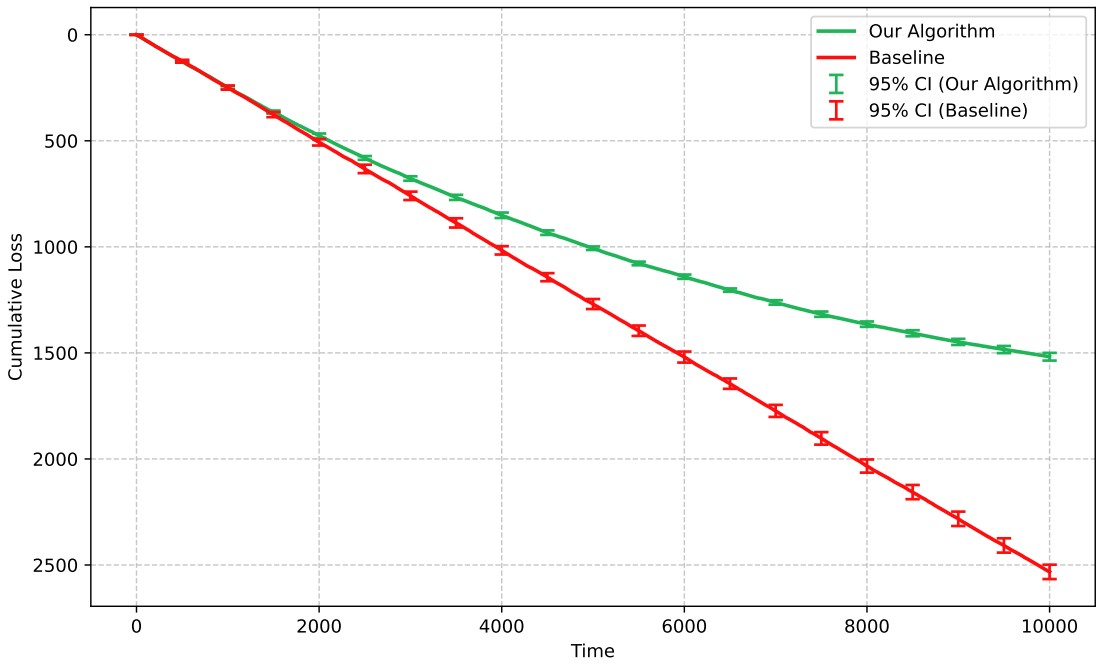

Figure 1. Comparison of average cumulative loss for Algorithm 1 and the OBPC baseline algorithm. The x-axis is time $t \in [T]$, and the y-axis is cumulative loss. The error bar indicates $\pm$ one standard deviation over 10 runs. Higher curves indicate better performance.

Taylor, J., Zemel, R., and Culotta, A. (eds.), *Advances in Neural Information Processing Systems*, volume 23. Curran Associates, Inc., 2010.

Kale, S., Lee, C., and Pál, D. Hardness of online sleeping combinatorial optimization problems. In Lee, D. D., Sugiyama, M., von Luxburg, U., Guyon, I., and Garnett, R. (eds.), *Advances in Neural Information Processing Systems 29: Annual Conference on Neural Information Processing Systems 2016, December 5-10, 2016, Barcelona, Spain*, pp. 2181–2189, 2016.

Kleinberg, R. D., Niculescu-Mizil, A., and Sharma, Y. Regret bounds for sleeping experts and bandits. *Machine Learning*, 80:245–272, 2010.

Lattimore, T. and Szepesvári, C. *Bandit Algorithms*. Cambridge University Press, 2020.

Li, L., Chu, W., Langford, J., and Schapire, R. E. A contextual-bandit approach to personalized news article recommendation. In *Proceedings of the 19th International Conference on World Wide Web*, WWW '10, pp. 661–670, New York, NY, USA, 2010. Association for Computing Machinery. ISBN 9781605587998. doi: 10.1145/1772690.1772758.

Liu, H., Wei, C.-Y., and Zimmert, J. Bypassing the simulator: Near-optimal adversarial linear contextual bandits. In Oh, A., Naumann, T., Globerson, A., Saenko, K., Hardt, M., and Levine, S. (eds.), *Advances in Neural Information Processing Systems*, volume 36, pp. 52086–52131. Curran Associates, Inc., 2023.

Neu, G. Explore no more: Improved high-probability regret bounds for non-stochastic bandits. In Cortes, C., Lawrence, N. D., Lee, D. D., Sugiyama, M., and Garnett, R. (eds.), *Advances in Neural Information Processing Systems 28: Annual Conference on Neural Information Processing Systems 2015, December 7-12, 2015, Montreal, Quebec, Canada*, pp. 3168–3176, 2015.

Neu, G. and Olkhovskaya, J. Efficient and robust algorithms for adversarial linear contextual bandits. In Abernethy, J. and Agarwal, S. (eds.), *Proceedings of Thirty Third Conference on Learning Theory*, volume 125 of *Proceedings of Machine Learning Research*, pp. 3049–3068. PMLR, 09–12 Jul 2020.

Neu, G. and Valko, M. Online combinatorial optimization with stochastic decision sets and adversarial losses. In Ghahramani, Z., Welling, M., Cortes, C., Lawrence, N.,

and Weinberger, K. (eds.), *Advances in Neural Information Processing Systems*, volume 27. Curran Associates, Inc., 2014.

Saha, A., Gaillard, P., and Valko, M. Improved sleeping bandits with stochastic actions sets and adversarial rewards. In *Proceedings of the 37th International Conference on Machine Learning*, ICML'20. JMLR.org, 2020.

Schneider, J. and Zimmert, J. Optimal cross-learning for contextual bandits with unknown context distributions. In Oh, A., Naumann, T., Globerson, A., Saenko, K., Hardt, M., and Levine, S. (eds.), *Advances in Neural Information Processing Systems*, volume 36, pp. 51862–51880. Curran Associates, Inc., 2023.

Syrgkanis, V., Luo, H., Krishnamurthy, A., and Schapire, R. E. Improved regret bounds for oracle-based adversarial contextual bandits. In Lee, D. D., Sugiyama, M., von Luxburg, U., Guyon, I., and Garnett, R. (eds.), *Advances in Neural Information Processing Systems 29: Annual Conference on Neural Information Processing Systems 2016, December 5-10, 2016, Barcelona, Spain*, pp. 3135–3143, 2016.

Villar, S. S., Bowden, J., and Wason, J. Multi-armed Bandit Models for the Optimal Design of Clinical Trials: Benefits and Challenges. *Statistical Science*, 30(2):199 – 215, 2015. doi: 10.1214/14-STS504.

Wang, Q., Yang, Z., Deng, X., and Kong, Y. Learning to bid in repeated first-price auctions with budgets. In Krause, A., Brunskill, E., Cho, K., Engelhardt, B., Sabato, S., and Scarlett, J. (eds.), *Proceedings of the 40th International Conference on Machine Learning*, volume 202 of *Proceedings of Machine Learning Research*, pp. 36494–36513. PMLR, 23–29 Jul 2023.

## A. Useful Lemmas

**Lemma 1** (Freedman's Inequality). *Fix any $\lambda > 0$ and $\delta \in (0, 1)$. Let $X_t$ be a random process with respect to a filtration $\mathcal{F}_t$ such that $\mu_t = \mathbb{E}\left[X_t \mid \mathcal{F}_{t-1}\right]$ and $V_t = \mathbb{E}\left[X_t^2 \mid \mathcal{F}_{t-1}\right]$, and satisfying $\lambda X_t \le 1$. Then, with probability at least $1 - \delta$, we have for all $t$,*

$$\sum_{s=1}^{t} X_s - \mu_s \le \lambda \sum_{s=1}^{t} V_s + \frac{\log(1/\delta)}{\lambda}.$$

The next lemma is about the following family of indicator functions.

**Definition 1.** *For each epoch $e$, we define the following two indicator functions:*

$$F_e \triangleq \mathbb{1}\left(\forall a, \left|\widehat{f}_e(a) - f_e(a)\right| \le 2\max\left\{\sqrt{\frac{f_e(a)\iota}{L}}, \frac{\iota}{L}\right\}\right)$$

*and*

$$L_e \triangleq \mathbb{1}\left(\max_{c\in[C], a\in[K]} \sum_{t\in\mathcal{T}_e} \widetilde{\ell}_{t,c}(a) \le L + \frac{\iota}{\gamma}\right).$$

*We further define the following indicator function:*

$$G = \prod_{e=1}^{T/L} F_e L_e.$$

**Lemma 2** (Lemma 6 and Lemma 7, Schneider & Zimmert (2023)). *For any epoch $e$, the event $F_e$ holds with probability at least $1 - 2K\exp(-\iota)$, and the event $L_e$ holds with probability at least $1 - K\exp(-\iota)$. Furthermore, the event $G$ holds with probability at least $1 - 3K(T/L)\exp(-\iota)$.*

**Lemma 3** (Lemma 8, Schneider & Zimmert (2023) ). *Let $\gamma \ge \frac{4\iota}{L}$, then under event $F_e$, we have that*

$$\frac{1}{2} \le \frac{f_e(a) + \gamma}{\widehat{f}_e(a) + \frac{3}{2}\gamma} \le 2.$$

The next lemma is about the following auxiliary probability vector.

**Definition 2.** *For each epoch $e$ and each time step $t \in \mathcal{T}_e$, we define*

$$\widetilde{p}_{t,c} \triangleq \arg\min_{p\in\Delta([K])} \left\langle p, \sum_{e'=1}^{e-1}\sum_{s\in\mathcal{T}_{e'}} \widehat{\ell}_{sc} + \sum_{t'\in\mathcal{T}_e, t'<t} \widetilde{\ell}_{t'c}\right\rangle - \eta^{-1}F(p)$$

*where $F(p) = \sum_{i=1}^{K} p_i \log(p_i)$ is the unnormalized negative entropy.*

It is easy to see that $\widetilde{p}_{t,c} \propto s_{e+1,c} \circ \exp\left(-\eta\sum_{t'\in\mathcal{T}_e, t'<t} \widetilde{\ell}_{t'c}\right)$.

**Lemma 4** (Lemma 9, Schneider & Zimmert (2023) ). *If $\gamma \ge \frac{4\iota}{L}$ and $\eta \le \frac{\log(2)}{5L}$, then under event $G$, we have for all $t \in \mathcal{T}_e, a \in [K], c \in [C]$ simultaneously*

$$2s_{e,c}(a) \ge p_{t,c}(a) \ge s_{e,c}(a)/2 \qquad \text{and} \qquad 2s_{e,c}(a) \ge \widetilde{p}_{t,c}(a) \ge s_{e,c}(a)/2.$$

*This implies that*

$$\mathbb{E}_{c\sim\nu}[p_{t,c}(a)] \le 4f_e(a) \qquad \text{and} \qquad \mathbb{E}_{c\sim\nu}[\widetilde{p}_{t,c}(a)] \le 4f_e(a).$$

*In addition, this implies that $q_t = p_t$ for all $t \in \mathcal{T}_e$.*

**Definition 3.** *We define $p_t(a) \triangleq \mathbb{E}_{c\sim\nu}[p_{t,c}(a)]$ and $\widetilde{p}_t(a) \triangleq \mathbb{E}_{c\sim\nu}[\widetilde{p}_{t,c}(a)]$ for each time step $t$ and each arm $a$.*

**Lemma 5** (Lemma 10, Schneider & Zimmert (2023) ). *If $\gamma \geq \frac{16\iota}{L}$ and $\exp(-\iota) \leq \frac{\gamma}{8K}$, then*

$$-\frac{\gamma}{f_e(a)} \leq \mathbb{E}\left[\frac{f_e(a) - \widehat{f}_e(a) - \frac{1}{2}\gamma}{\widehat{f}_e(a) + \frac{3}{2}\gamma} F_e \Big| \mathcal{H}_{e-1}\right] \leq 0.$$

**Lemma 6.** *For any $\eta \leq \frac{\gamma}{2(2L\gamma+\iota)}, \gamma \geq \frac{16\iota}{L}, \iota \geq \log(8K/\gamma)$, we have*

$$\sum_{t \in \mathcal{T}^\ell} \sum_c \Pr(c) \left\langle p_{t,c} - \widetilde{p}_{t,c}, \widetilde{\ell}_{t,c} - \widehat{\ell}_{t,c} \right\rangle G \leq \frac{98KT\iota}{L} + \frac{\gamma^2 LKT}{\iota}.$$

*Proof.* The proof of Lemma 6 is contained in the analysis of the **bias**$_3$ term in Schneider & Zimmert (2023). □

**Lemma 7.** *Decomposing all time steps into consecutive pairs, specifically, decomposing $\{1, 2, \ldots, T\}$ into $\{(1, 2), (3, 4), (5, 6), \ldots, (t-1, t), \ldots, (T-1, T)\}$. Constructing a surrogate loss sequence $\{\widetilde{\ell}_s\}_{s=1}^{\frac{T}{2}}$ such that for each surrogate time step $s$ the loss vector $\widetilde{\ell}_s$ is uniformly sampled from the pair of true loss vector $(\ell_{2s-1}, \ell_{2s})$. Denote the time step sampled from the pair $(2s-1, 2s)$ as $s_\ell$. For any constant $\delta \in (0, 1)$ and any bandit algorithm such that in each pair of time steps $(t-1, t)$, the algorithm takes actions $a_{t-1}$ and $a_t$ from the same distribution $p_{t-1} = p_t$, we have*

$$\sum_{t=1}^T (\ell_{t,c_t}(a_t) - \ell_{t,c_t}(\pi_{c_t})) \leq 2\sum_{s=1}^{\frac{T}{2}} \left(\widetilde{\ell}_{s,c_{s_\ell}}(a_{s_\ell}) - \widetilde{\ell}_{s,c_{s_\ell}}(\pi_{c_{s_\ell}})\right) + 2\sqrt{T\log(\frac{1}{\delta})}$$

*with probability at least $1 - \delta$.*

*Proof of Lemma 7.* Consider the sequence of random variable $\{Y_s\}_{s=1}^{\frac{T}{2}}$ such that

$$\begin{aligned}
Y_s =& \ell_{2s-1,c_{2s-1}}(a_{2s-1}) - \ell_{2s-1,c_{2s-1}}(\pi_{c_{2s-1}}) \\
&+ \ell_{2s,c_{2s}}(a_{2s}) - \ell_{2s,c_{2s}}(\pi_{c_{2s}}) \\
&- 2\left(\widetilde{\ell}_{s,c_{s_\ell}}(a_{s_\ell}) - \widetilde{\ell}_{s,c_{s_\ell}}(\pi_{c_{s_\ell}})\right).
\end{aligned}$$

Consider the filtration $\{\widetilde{H}_s\}_{s=1}^{\frac{T}{2}}$ such that for each $s$ the $\sigma$-field $\widetilde{H}_s$ is generated by the randomness within $c_t$ and $a_t$ for $t \leq 2s$ and the randomness within sampling from pair $(2\tau-1, 2\tau)$ for each $\tau \leq s$. It is easy to see that the sequence $\{Y_s\}_{s=1}^{\frac{T}{2}}$ forms a martingale difference sequence adapted to the filtration $\{\widetilde{H}_s\}_{s=1}^{\frac{T}{2}}$. Moreover, it is also easy to see that $|Y_s| \leq 2$. Using Azuma-Hoeffding's inequality, for any constant $\delta \in (0, 1)$, we have

$$\sum_{s=1}^{\frac{T}{2}} Y_s \leq 2\sqrt{T\log(\frac{1}{\delta})}$$

with probability at least $1 - \delta$. This completes the proof of the lemma. □

## B. Detailed Proof of Theorem 1

### B.1. Decomposition

As we mentioned in Section 4, we decompose the regret as

$$\text{Reg}(\pi) = \underbrace{\sum_{t=1}^{T} \ell_{t,c_t}(a_t) - \ell_{t,c_t}(\pi_{c_t}) - 2 \sum_{t \in \mathcal{T}^\ell} \ell_{t,c_t}(a_t) - \ell_{t,c_t}(\pi_{c_t})}_{\textbf{bias}_1}$$

$$+ 2 \underbrace{\sum_{t \in \mathcal{T}^\ell} \left( \ell_{t,c_t}(a_t) - \ell_{t,c_t}(\pi_{c_t}) - \sum_c \Pr(c)\langle p_{t,c} - \pi_c, \ell_{t,c}\rangle \right)}_{\textbf{bias}_2}$$

$$+ 2 \underbrace{\sum_{t \in \mathcal{T}^\ell} \sum_c \Pr(c)\langle p_{t,c} - \pi_c, \widehat{\ell}_{t,c}\rangle}_{\textbf{ftrl}} + 2 \underbrace{\sum_{t \in \mathcal{T}^\ell} \sum_c \Pr(c) \left\langle p_{t,c}, \ell_{t,c} - \widetilde{\ell}_{t,c}\right\rangle}_{\textbf{bias}_3}$$

$$+ 2 \underbrace{\sum_{t \in \mathcal{T}^\ell} \sum_c \Pr(c) \left\langle p_{t,c}, \widetilde{\ell}_{t,c} - \widehat{\ell}_{t,c}\right\rangle}_{\textbf{bias}_4} + 2 \underbrace{\sum_{t \in \mathcal{T}^\ell} \sum_c \Pr(c) \left\langle \pi_c, \widehat{\ell}_{t,c} - \ell_{t,c}\right\rangle}_{\textbf{bias}_5}.$$

We bound these terms one by one.

The **bias**$_1$, **bias**$_2$, **ftrl**, and **bias**$_3$ terms are not hard to bound. The terms of interest to bound are **bias**$_4$ ans **bias**$_5$. We first bound these two terms. In these two terms, the **bias**$_5$ term is the one easier to bound. We first bound **bias**$_5$ to provide some intuition for our readers.

### B.2. Upper Bounding bias$_5$

We first bound the fifth term $\sum_{t \in \mathcal{T}^\ell} \sum_c \Pr(c) \left\langle \pi_c, \widehat{\ell}_{t,c} - \ell_{t,c}\right\rangle$. We decompose the fifth term into two components:

$$\sum_{t \in \mathcal{T}^\ell} \sum_c \Pr(c) \left\langle \pi_c, \widehat{\ell}_{t,c} - \ell_{t,c}\right\rangle$$

$$= \sum_{t \in \mathcal{T}^\ell} \sum_c \Pr(c) \left( \widehat{\ell}_{t,c}(\pi_c) - \ell_{t,c}(\pi_c)\right)$$

$$= \sum_{t \in \mathcal{T}^\ell} \sum_c \Pr(c) \left( \mathbb{E}\left[\widehat{\ell}_{t,c}(\pi_c)\big|\, \mathcal{H}_{t-1}\right] - \ell_{t,c}(\pi_c)\right)$$

$$+ \sum_{t \in \mathcal{T}^\ell} \sum_c \Pr(c) \left( \widehat{\ell}_{t,c}(\pi_c) - \mathbb{E}\left[\widehat{\ell}_{t,c}(\pi_c)\big|\, \mathcal{H}_{t-1}\right]\right).$$

We bound these two components separately.

For each epoch $e$, we define a random variable

$$\text{Bias5}_e \triangleq \sum_{t \in \mathcal{T}_e^\ell} \sum_c \Pr(c) \left( \mathbb{E}\left[\widehat{\ell}_{t,c}(\pi_c)\big|\, \mathcal{H}_{t-1}\right] - \ell_{t,c}(\pi_c)\right).$$

We rewrite the term $\sum_{t \in \mathcal{T}^\ell} \sum_c \Pr(c) \left( \mathbb{E}\left[\widehat{\ell}_{t,c}(\pi_c)\big|\, \mathcal{H}_{t-1}\right] - \ell_{t,c}(\pi_c)\right)$ as $\sum_{e=1}^{T/L} \text{Bias5}_e$. Recall our key observation: different epochs are only weakly dependent on each other. We bound the summation over epochs $\sum_{e=1}^{T/L} \text{Bias5}_e$ by leveraging the weak dependence structure between $\{\text{Bias5}_e\}_e$.

The sequence of random variables $\{\text{Bias5}_e\}_e$ has the following properties:

1. For each epoch $e$, the random variable $\text{Bias5}_e$ is measurable under $\sigma$-algebra $\mathcal{H}_e$.

2. For each epoch $e$, we have[4]

$$\mathbb{E}\left[\text{Bias5}_e \cdot F_e \,\middle|\, \mathcal{H}_{e-1}\right]$$
$$= \sum_c \Pr(c) \sum_{t \in \mathcal{T}_e^\ell} \ell_{t,c}(\pi_c) \,\mathbb{E}\left[\left(\frac{f_e(\pi_c)}{\widehat{f}_e(\pi_c) + \frac{3}{2}\gamma} - 1\right) F_e \,\middle|\, \mathcal{H}_{e-1}\right].$$

We further have

$$\mathbb{E}\left[\left(\frac{f_e(\pi_c)}{\widehat{f}_e(\pi_c) + \frac{3}{2}\gamma} - 1\right) F_e \,\middle|\, \mathcal{H}_{e-1}\right]$$
$$= \mathbb{E}\left[\left(\frac{f_e(\pi_c)}{\widehat{f}_e(\pi_c) + \frac{3}{2}\gamma} - \frac{f_e(\pi_c)}{f_e(\pi_c) + \gamma} + \frac{f_e(\pi_c)}{f_e(\pi_c) + \gamma} - 1\right) F_e \,\middle|\, \mathcal{H}_{e-1}\right]$$
$$= \mathbb{E}\left[\frac{f_e(\pi_c)}{f_e(\pi_c) + \gamma} \frac{\left(f_e(\pi_c) - \widehat{f}_e(\pi_c) - \frac{1}{2}\gamma\right)}{\widehat{f}_e(\pi_c) + \frac{3}{2}\gamma} F_e \,\middle|\, \mathcal{H}_{e-1}\right] - \frac{\gamma}{f_e(\pi_c) + \gamma} \mathbb{E}\left[F_e \,\middle|\, \mathcal{H}_{e-1}\right]$$
$$\leq -\frac{\gamma}{f_e(\pi_c) + \gamma} \mathbb{E}\left[F_e \,\middle|\, \mathcal{H}_{e-1}\right] \qquad\qquad\qquad\qquad (\text{Lemma } 5)$$
$$\leq -\frac{\gamma}{f_e(\pi_c) + \gamma} \left(1 - 2K \exp(-\iota)\right). \qquad\qquad\qquad (\text{Lemma } 2)$$

Thus we have

$$\mathbb{E}\left[\text{Bias5}_e \cdot F_e \,\middle|\, \mathcal{H}_{e-1}\right]$$
$$\leq -\sum_c \Pr(c) \sum_{t \in \mathcal{T}_e^\ell} \ell_{t,c}(\pi_c) \frac{\gamma}{f_e(\pi_c) + \gamma} \left(1 - 2K \exp(-\iota)\right).$$

3. For each epoch $e$, we have

$$\text{Bias5}_e \, F_e \leq \sum_c \Pr(c) \sum_{t \in \mathcal{T}_e^\ell} \mathbb{E}\left[\widehat{\ell}_{t,c}(\pi_c) \,\middle|\, \mathcal{H}_{t-1}\right] F_e$$
$$= \sum_c \Pr(c) \sum_{t \in \mathcal{T}_e^\ell} \ell_{t,c}(\pi_c) \frac{f_e(\pi_c)}{\widehat{f}_e(\pi_c) + \frac{3}{2}\gamma} F_e$$
$$\leq \sum_c \Pr(c) \sum_{t \in \mathcal{T}_e^\ell} \ell_{t,c}(\pi_c) \frac{2f_e(\pi_c)}{f_e(\pi_c) + \gamma} \qquad\qquad\qquad (\text{Lemma } 3)$$
$$\leq 2L = \frac{32\iota}{\gamma}.$$

---

[4]Readers familiar with Schneider & Zimmert (2023) may wonder why we do not directly consider $\text{Bias5}_e \, G$ but consider $\text{Bias5}_e \, F_e$ instead. This is because there is a small flaw in the argument of Schneider & Zimmert (2023). Schneider & Zimmert (2023) essentially argues that $\mathbb{E}[\text{Bias5}_e \, G \,|\, \mathcal{H}_{e-1}] = \mathbb{E}[\text{Bias5}_e \,|\, \mathcal{H}_{e-1}] \mathbb{E}[G \,|\, \mathcal{H}_{e-1}]$. However, this equality may not hold since the indicator $G$ depends on $\text{Bias5}_e$ and these two terms are not conditionally independent given $\mathcal{H}_{e-1}$. This is why we consider $\text{Bias5}_e \, F_e$ here instead.

4. For each epoch $e$, we have

$$\mathbb{E}\left[(\text{Bias5}_e\, F_e)^2 \,\middle|\, \mathcal{H}_{e-1}\right]$$

$$= \mathbb{E}\left[\left(\sum_c \Pr(c) \sum_{t\in\mathcal{T}_e^\ell} \ell_{t,c}(\pi_c)\frac{f_e(\pi_c) - \widehat{f}_e(\pi_c) - \frac{3}{2}\gamma}{\widehat{f}_e(\pi_c) + \frac{3}{2}\gamma}F_e\right)^2 \,\middle|\, \mathcal{H}_{e-1}\right]$$

$$\leq \sum_c \Pr(c)\, \mathbb{E}\left[\left(\sum_{t\in\mathcal{T}_e^\ell} \ell_{t,c}(\pi_c)\frac{f_e(\pi_c) - \widehat{f}_e(\pi_c) - \frac{3}{2}\gamma}{\widehat{f}_e(\pi_c) + \frac{3}{2}\gamma}F_e\right)^2 \,\middle|\, \mathcal{H}_{e-1}\right]$$

$$= \sum_c \Pr(c)(\sum_{t\in\mathcal{T}_e^\ell} \ell_{t,c})^2\, \mathbb{E}\left[\left(\frac{f_e(\pi_c) - \widehat{f}_e(\pi_c) - \frac{3}{2}\gamma}{\widehat{f}_e(\pi_c) + \frac{3}{2}\gamma}F_e\right)^2 \,\middle|\, \mathcal{H}_{e-1}\right]$$

$$\leq L\sum_c \Pr(c) \sum_{t\in\mathcal{T}_e^\ell} \ell_{t,c}\, \mathbb{E}\left[\left(\frac{f_e(\pi_c) - \widehat{f}_e(\pi_c) - \frac{3}{2}\gamma}{\widehat{f}_e(\pi_c) + \frac{3}{2}\gamma}F_e\right)^2 \,\middle|\, \mathcal{H}_{e-1}\right]$$

$$\leq 4L\sum_c \Pr(c) \sum_{t\in\mathcal{T}_e^\ell} \ell_{t,c}\, \mathbb{E}\left[\left(\frac{f_e(\pi_c) - \widehat{f}_e(\pi_c) - \frac{3}{2}\gamma}{f_e(\pi_c) + \gamma}F_e\right)^2 \,\middle|\, \mathcal{H}_{e-1}\right] \qquad \text{(Lemma 3)}$$

$$\leq \sum_c \Pr(c)\frac{4L}{(f_e(\pi_c) + \gamma)^2} \sum_{t\in\mathcal{T}_e^\ell} \ell_{t,c}(\pi_c)\, \mathbb{E}\left[\left(f_e(\pi_c) - \widehat{f}_e(\pi_c) - \frac{3}{2}\gamma\right)^2 \,\middle|\, \mathcal{H}_{e-1}\right]$$

$$= \sum_c \Pr(c)\frac{4L}{(f_e(\pi_c) + \gamma)^2} \sum_{t\in\mathcal{T}_e^\ell} \ell_{t,c}(\pi_c)\left(\mathbb{E}\left[\left(f_e(\pi_c) - \widehat{f}_e(\pi_c)\right)^2 \,\middle|\, \mathcal{H}_{e-1}\right] + \frac{9}{4}\gamma^2\right)$$

$$\leq \sum_c \Pr(c)\frac{4L}{(f_e(\pi_c) + \gamma)^2} \sum_{t\in\mathcal{T}_e^\ell}\sum_{t\in\mathcal{T}_e^\ell} \ell_{t,c}(\pi_c)\left(\frac{f_e(\pi_c)}{L} + \frac{9}{4}\gamma^2\right)$$

$$\leq 4\sum_c \Pr(c) \sum_{t\in\mathcal{T}_e^\ell} \ell_{t,c}(\pi_c)\left(\frac{1}{f_e(\pi_c) + \gamma} + \frac{9L\gamma}{4(f_e(\pi_c) + \frac{3}{2}\gamma)}\right)$$

$$\leq 4\sum_c \Pr(c) \sum_{t\in\mathcal{T}_e^\ell} \ell_{t,c}(\pi_c)\frac{36\iota}{f_e(\pi_c) + \gamma}.$$

Given these properties, we use Freedman's inequality to get that for any $0 < \lambda < \frac{\gamma}{32\iota}$, with probability at least $1 - \delta$, we have

$$\sum_e \text{Bias5}_e\, F_e - \mathbb{E}[\text{Bias5}_e\, F_e \,|\, \mathcal{H}_{e-1}] \leq 4\lambda \sum_c \Pr(c) \sum_{t\in\mathcal{T}_e^\ell} \ell_{t,c}(\pi_c)\frac{36\iota}{f_e(\pi_c) + \frac{3}{2}\gamma} + \frac{\log(1/\delta)}{\lambda}.$$

We further have that event $\{\sum_e \text{Bias5}_e\, F_e = \sum_e \text{Bias5}_e\}$ holds if event $G$ holds. Combining these two facts, we get that the inequality

$$\sum_e \text{Bias5}_e - \mathbb{E}[\text{Bias5}_e\, F_e \,|\, \mathcal{H}_{e-1}] \leq 4\lambda \sum_c \Pr(c) \sum_{t\in\mathcal{T}_e^\ell} \ell_{t,c}(\pi_c)\frac{36\iota}{f_e(\pi_c) + \gamma} + \frac{\log(1/\delta)}{\lambda}$$

holds with probability at least $\Pr(G) - \delta$.

We now bound the second component

$$\sum_{t\in\mathcal{T}^\ell}\sum_c \Pr(c)\left(\widehat{\ell}_{t,c}(\pi_c) - \mathbb{E}\left[\widehat{\ell}_{t,c}(\pi_c) \,\middle|\, \mathcal{H}_{t-1}\right]\right).$$

The second term $\sum_{t\in\mathcal{T}^\ell}\sum_c \Pr(c)\left(\widehat{\ell}_{t,c}(\pi_c) - \mathbb{E}\left[\widehat{\ell}_{t,c}(\pi_c) \,\middle|\, \mathcal{H}_{t-1}\right]\right)$ has the following properties:

1. The sequence of random variables $\left\{ \sum_c \Pr(c) \left( \widehat{\ell}_{t,c}(\pi_c) - \mathbb{E}\left[ \widehat{\ell}_{t,c}(\pi_c) \middle| \mathcal{H}_{t-1} \right] \right) \right\}_{t \in \mathcal{T}^\ell}$ forms a martingale difference sequence with respect to the tfiltration $\{\mathcal{H}_t\}_t$.

2. Each random variable $\sum_c \Pr(c) \left( \widehat{\ell}_{t,c}(\pi_c) - \mathbb{E}\left[ \widehat{\ell}_{t,c}(\pi_c) \middle| \mathcal{H}_{t-1} \right] \right)$ satisfies $\left| \sum_c \Pr(c) \left( \widehat{\ell}_{t,c}(\pi_c) - \mathbb{E}\left[ \widehat{\ell}_{t,c}(\pi_c) \middle| \mathcal{H}_{t-1} \right] \right) \right| \leq \frac{1}{\gamma}$.

3. Each random variable $\sum_c \Pr(c) \left( \widehat{\ell}_{t,c}(\pi_c) - \mathbb{E}\left[ \widehat{\ell}_{t,c}(\pi_c) \middle| \mathcal{H}_{t-1} \right] \right)$ satisfies

$$
\mathrm{Var}\left[ \sum_c \Pr(c)\widehat{\ell}_{t,c}(\pi_c) \middle| \mathcal{H}_{t-1} \right] \leq \sum_c \Pr(c) \, \mathrm{Var}\left[ \widehat{\ell}_{t,c}(\pi_c) \middle| \mathcal{H}_{t-1} \right]
$$
$$
= \sum_c \Pr(c) \frac{f_e(\pi_c) - f_e^2(\pi_c)}{\left( \widehat{f}_e(\pi_c) + \frac{3}{2}\gamma \right)^2} l_{t,c}(\pi_c)^2
$$
$$
\leq \sum_c \Pr(c) \frac{f_e(\pi_c)}{\left( \widehat{f}_e(\pi_c) + \frac{3}{2}\gamma \right)^2} l_{t,c}(\pi_c).
$$

Applying Freedman's inequality to the sequence of random variables $\left\{ \sum_c \Pr(c) \left( \widehat{\ell}_{t,c}(\pi_c) - \mathbb{E}\left[ \widehat{\ell}_{t,c}(\pi_c) \middle| \mathcal{H}_{t-1} \right] \right) \right\}_{t \in \mathcal{T}^\ell}$, we get that for each $\delta \in (0,1)$ and each $0 < \lambda < \gamma$, with probability at least $1 - \delta$, we have

$$
\sum_{t \in \mathcal{T}^\ell} \sum_c \Pr(c) \left( \widehat{\ell}_{t,c}(\pi_c) - \mathbb{E}\left[ \widehat{\ell}_{t,c}(\pi_c) \middle| \mathcal{H}_{t-1} \right] \right)
$$
$$
\leq \lambda \sum_{t \in \mathcal{T}^\ell} \sum_c \Pr(c) \frac{f_e(\pi_c)}{\left( \widehat{f}_e(\pi_c) + \frac{3}{2}\gamma \right)^2} l_{t,c}(\pi_c) + \frac{1}{\lambda} \log(\frac{1}{\delta}).
$$

By assuming that event $G$ holds, we further get that with probability at least $\Pr(G) - \delta$, the following inequality holds:

$$
\sum_{t \in \mathcal{T}^\ell} \sum_c \Pr(c) \left( \widehat{\ell}_{t,c}(\pi_c) - \mathbb{E}\left[ \widehat{\ell}_{t,c}(\pi_c) \middle| \mathcal{H}_{t-1} \right] \right)
$$
$$
\leq 4\lambda \sum_{t \in \mathcal{T}^\ell} \sum_c \Pr(c) \frac{f_e(\pi_c)}{\left( f_e(\pi_c) + \gamma \right)^2} l_{t,c}(\pi_c) + \frac{1}{\lambda} \log(\frac{1}{\delta}). \qquad \text{(Lemma 3)}
$$

Combining all previous inequalities, we get that for any $0 < \lambda_1 < \frac{\gamma}{32\iota}$ and $0 < \lambda_2 < \gamma$, the following inequality holds with

probability at least $\Pr(G) - 2\delta$:

$$\sum_{t \in \mathcal{T}^\ell} \sum_c \Pr(c) \left\langle \pi_c, \widehat{\ell}_{t,c} - \ell_{t,c} \right\rangle$$

$$= \sum_{t \in \mathcal{T}^\ell} \sum_c \Pr(c) \left( \mathbb{E}\left[ \widehat{\ell}_{t,c}(\pi_c) \middle| \mathcal{H}_{t-1} \right] - \ell_{t,c}(\pi_c) \right)$$

$$+ \sum_{t \in \mathcal{T}^\ell} \sum_c \Pr(c) \left( \widehat{\ell}_{t,c}(\pi_c) - \mathbb{E}\left[ \widehat{\ell}_{t,c}(\pi_c) \middle| \mathcal{H}_{t-1} \right] \right)$$

$$= \sum_e \text{Bias5}_e - \mathbb{E}[\text{Bias5}_e \, F_e \middle| \mathcal{H}_{e-1}]$$

$$+ \sum_e \mathbb{E}\left[ \text{Bias5}_e \, F_e \middle| \mathcal{H}_{e-1} \right]$$

$$+ \sum_{t \in \mathcal{T}^\ell} \sum_c \Pr(c) \left( \widehat{\ell}_{t,c}(\pi_c) - \mathbb{E}\left[ \widehat{\ell}_{t,c}(\pi_c) \middle| \mathcal{H}_{t-1} \right] \right)$$

$$\leq 4\lambda_1 \sum_c \Pr(c) \sum_{t \in \mathcal{T}^\ell} \ell_{t,c}(\pi_c) \frac{36\iota}{f_e(\pi_c) + \gamma} + \frac{\log(1/\delta)}{\lambda_1}$$

$$- \sum_c \Pr(c) \sum_{t \in \mathcal{T}^\ell} \ell_{t,c}(\pi_c) \frac{\gamma}{f_e(\pi_c) + \gamma} \left( 1 - 2K \exp(-\iota) \right)$$

$$+ 4\lambda_2 \sum_{t \in \mathcal{T}^\ell} \sum_c \Pr(c) \frac{f_e(\pi_c)}{(f_e(\pi_c) + \gamma)^2} l_{t,c}(\pi_c) + \frac{1}{\lambda_2} \log(\frac{1}{\delta}).$$

Note that in the previous analysis, we combined two good events each happening with probability at least $\Pr(G) - \delta$. The combined good event happens with probability $\Pr(G) - 2\delta$ rather than the vanilla union bound $1 - 2(1 - \Pr(G) + \delta)$. This is because, in both events, the $\Pr(G)$ term comes from assuming event $G$ happens. Thus, in the combined event, we can simply assume event $G$ happens and count the corresponding bad event $G^c$ only once. We will use this small trick repeatedly in the following analysis.

We pick $\lambda_1 = \frac{\gamma}{8 \cdot 36\iota}$ and $\lambda_2 = \frac{\gamma}{8}$ to get that the following inequality holds with probability at least $\Pr(G) - 2\delta$:

$$\sum_{t \in \mathcal{T}^\ell} \sum_c \Pr(c) \left\langle \pi_c, \widehat{\ell}_{t,c} - \ell_{t,c} \right\rangle$$

$$\leq 4\lambda_1 \sum_c \Pr(c) \sum_{t \in \mathcal{T}^\ell} \ell_{t,c}(\pi_c) \frac{36\iota}{f_e(\pi_c) + \gamma} + \frac{\log(1/\delta)}{\lambda_1}$$

$$- \sum_c \Pr(c) \sum_{t \in \mathcal{T}^\ell} \ell_{t,c}(\pi_c) \frac{\gamma}{f_e(\pi_c) + \gamma} \left( 1 - 2K \exp(-\iota) \right)$$

$$+ 4\lambda_2 \sum_{t \in \mathcal{T}^\ell} \sum_c \Pr(c) \frac{f_e(\pi_c)}{(f_e(\pi_c) + \gamma)^2} l_{t,c}(\pi_c) + \frac{1}{\lambda_2} \log(\frac{1}{\delta})$$

$$= \left( \frac{8 \cdot 36\iota}{\gamma} + \frac{8}{\gamma} \right) \log(1/\delta) + \sum_c \Pr(c) \sum_{t \in \mathcal{T}^\ell} \ell_{t,c}(\pi_c) \frac{\gamma}{f_e(\pi_c) + \gamma} 2K \exp(-\iota)$$

$$\leq \left( \frac{8 \cdot 36\iota}{\gamma} + \frac{8}{\gamma} \right) \log(1/\delta) + KT \exp(-\iota).$$

**B.3. Upper Bounding** $\text{bias}_4$

We then bound the forth term $\sum_{t \in \mathcal{T}^\ell} \sum_c \Pr(c) \left\langle p_{t,c}, \widetilde{\ell}_{t,c} - \widehat{\ell}_{t,c} \right\rangle$. Similar to the previous analysis, we decompose it as follows:

$$
\sum_{t \in \mathcal{T}^\ell} \sum_c \Pr(c) \left\langle p_{t,c}, \widetilde{\ell}_{t,c} - \widehat{\ell}_{t,c} \right\rangle
$$
$$
= \sum_{t \in \mathcal{T}^\ell} \sum_c \Pr(c) \left\langle p_{t,c}, \mathbb{E}\left[\widetilde{\ell}_{t,c} - \widehat{\ell}_{t,c} \middle| \mathcal{H}_{t-1}\right] \right\rangle
$$
$$
+ \sum_{t \in \mathcal{T}^\ell} \sum_c \Pr(c) \left\langle p_{t,c}, \widetilde{\ell}_{t,c} - \widehat{\ell}_{t,c} - \mathbb{E}\left[\widetilde{\ell}_{t,c} - \widehat{\ell}_{t,c} \middle| \mathcal{H}_{t-1}\right] \right\rangle.
$$

We bound these two components separately.

Similar to the previous anlaysis, we decompose the first component

$$
\sum_{t \in \mathcal{T}^\ell} \sum_c \Pr(c) \left\langle p_{t,c}, \mathbb{E}\left[\widetilde{\ell}_{t,c} - \widehat{\ell}_{t,c} \middle| \mathcal{H}_{t-1}\right] \right\rangle
$$

as

$$
\sum_e \sum_{t \in \mathcal{T}_e^\ell} \sum_c \Pr(c) \left\langle p_{t,c}, \mathbb{E}\left[\widetilde{\ell}_{t,c} - \widehat{\ell}_{t,c} \middle| \mathcal{H}_{t-1}\right] \right\rangle.
$$

For each epoch $e$ we define a random variable

$$
\text{Bias4}_e \triangleq \sum_{t \in \mathcal{T}_e^\ell} \sum_c \Pr(c) \left\langle p_{t,c}, \mathbb{E}\left[\widetilde{\ell}_{t,c} - \widehat{\ell}_{t,c} \middle| \mathcal{H}_{t-1}\right] \right\rangle.
$$

We need to bound $\sum_e \text{Bias4}_e$.

We decompose $\sum_e \text{Bias4}_e$ as $\sum_e \text{Bias4}\, F_e L_e + \sum_e \text{Bias4}_e (1 - F_e L_e)$. As usual we have that $\sum_e \text{Bias4}_e(1 - F_e L_e) = 0$ whenever event $G$ holds. Thus we can focus on bounding $\sum_e \text{Bias4}_e\, F_e L_e$. Firstly we bound

$$
\sum_e \mathbb{E}\left[\text{Bias4}_e\, F_e L_e \middle| \mathcal{H}_{e-1}\right].
$$

We have

$$
\sum_e \mathbb{E}\left[\text{Bias4}_e\, F_e L_e \middle| \mathcal{H}_{e-1}\right]
$$
$$
= \sum_e \mathbb{E}\left[\sum_{t \in \mathcal{T}_e^\ell} \sum_c \Pr(c) \left\langle \widetilde{p}_{t,c}, \mathbb{E}\left[\widetilde{\ell}_{t,c} - \widehat{\ell}_{t,c} \middle| \mathcal{H}_{t-1}\right] \right\rangle F_e L_e \middle| \mathcal{H}_{e-1}\right]
$$
$$
+ \sum_e \mathbb{E}\left[\sum_{t \in \mathcal{T}_e^\ell} \sum_c \Pr(c) \left\langle p_{t,c} - \widetilde{p}_{t,c}, \mathbb{E}\left[\widetilde{\ell}_{t,c} - \widehat{\ell}_{t,c} \middle| \mathcal{H}_{t-1}\right] \right\rangle F_e L_e \middle| \mathcal{H}_{e-1}\right].
$$

By Lemma 6, the latter term

$$
\sum_e \mathbb{E}\left[\sum_{t \in \mathcal{T}_e^\ell} \sum_c \Pr(c) \left\langle p_{t,c} - \widetilde{p}_{t,c}, \mathbb{E}\left[\widetilde{\ell}_{t,c} - \widehat{\ell}_{t,c} \middle| \mathcal{H}_{t-1}\right] \right\rangle F_e L_e \middle| \mathcal{H}_{e-1}\right]
$$

is bounded by $\frac{98KT\iota}{L} + \frac{\gamma^2 LKT}{\iota}$. Furthermore, condition on $\mathcal{H}_{e-1}$, the indicator function $F_e$ is effected only by randomness

within time steps $t \in \mathcal{T}_e^{\mathrm{f}}$, thus the indicator function $F_e$ is conditional independent with the probability vector $\widetilde{p}_{t,c}$. We have

$$\sum_e \mathbb{E}\left[\sum_{t \in \mathcal{T}_e^\ell}\sum_c \mathrm{Pr}(c)\left\langle \widetilde{p}_{t,c}, \mathbb{E}\left[\widetilde{\ell}_{t,c} - \widehat{\ell}_{t,c}\middle| \mathcal{H}_{t-1}\right]\right\rangle F_e\middle| \mathcal{H}_{e-1}\right]$$

$$= \sum_e \sum_{t \in \mathcal{T}_e^\ell}\sum_c \mathrm{Pr}(c)\left\langle \mathbb{E}\left[\widetilde{p}_{t,c}\middle| \mathcal{H}_{e-1}\right], \mathbb{E}\left[(\widetilde{\ell}_{t,c} - \widehat{\ell}_{t,c})F_e\middle| \mathcal{H}_{e-1}\right]\right\rangle$$

$$\leq \sum_e \sum_{t \in \mathcal{T}_e^\ell}\sum_c \mathrm{Pr}(c)\sum_a \frac{\widetilde{p}_{t,c}(a)\gamma}{f_e(a)}. \qquad \text{(Lemma 5)}$$

By Lemma 4, whenever event $G$ holds, the ratio $\frac{\widetilde{p}_{t,c}(a)}{f_e(a)} \leq 4$. Thus we have

$$\sum_e \mathbb{E}\left[\sum_{t \in \mathcal{T}_e^\ell}\sum_c \mathrm{Pr}(c)\left\langle \widetilde{p}_{t,c}, \mathbb{E}\left[\widetilde{\ell}_{t,c} - \widehat{\ell}_{t,c}\middle| \mathcal{H}_{t-1}\right]\right\rangle F_e\middle| \mathcal{H}_{e-1}\right] \leq 4\gamma KT$$

whenever event $G$ holds.

We further have $\left|\left\langle \widetilde{p}_{t,c}, \mathbb{E}\left[\widetilde{\ell}_{t,c} - \widehat{\ell}_{t,c}\middle| \mathcal{H}_{t-1}\right]\right\rangle\right| \leq \frac{1}{\gamma}$. Thus we have

$$\sum_e \mathbb{E}\left[\sum_{t \in \mathcal{T}_e^\ell}\sum_c \mathrm{Pr}(c)\left\langle \widetilde{p}_{t,c}, \mathbb{E}\left[\widetilde{\ell}_{t,c} - \widehat{\ell}_{t,c}\middle| \mathcal{H}_{t-1}\right]\right\rangle F_e(L_e - 1)\middle| \mathcal{H}_{e-1}\right]$$

$$\leq \sum_e \mathbb{E}\left[\sum_{t \in \mathcal{T}_e^\ell}\sum_c \mathrm{Pr}(c)\frac{1}{\gamma}F_e(L_e - 1)\middle| \mathcal{H}_{e-1}\right]$$

$$\leq \frac{1}{\gamma}\sum_e \mathbb{E}\left[\sum_{t \in \mathcal{T}_e^\ell}\sum_c \mathrm{Pr}(c)\left|L_e - 1\right|\middle| \mathcal{H}_{e-1}\right]$$

$$\leq \frac{K\exp(-\iota)T}{\gamma}. \qquad \text{(Lemma 2)}$$

Thus we have

$$\sum_e \mathbb{E}\left[\sum_{t \in \mathcal{T}_e^\ell}\sum_c \mathrm{Pr}(c)\left\langle \widetilde{p}_{t,c}, \mathbb{E}\left[\widetilde{\ell}_{t,c} - \widehat{\ell}_{t,c}\middle| \mathcal{H}_{t-1}\right]\right\rangle F_e L_e\middle| \mathcal{H}_{e-1}\right]$$

$$\leq 4\gamma KT + \frac{K\exp(-\iota)T}{\gamma}$$

whenever event $G$ holds.

Thus we have

$$\sum_e \mathbb{E}\left[\mathrm{Bias4}_e\, F_e L_e\middle| \mathcal{H}_{e-1}\right] \leq 4K\gamma T + \frac{K\exp(-\iota)T}{\gamma} + \frac{98KT\iota}{L} + \frac{\gamma^2 LKT}{\iota}$$

whenever event $G$ holds.

We then only need to bound the concentration term

$$\sum_e \mathrm{Bias4}_e\, F_e L_e - \mathbb{E}\left[\mathrm{Bias4}_e\, F_e L_e\middle| \mathcal{H}_{e-1}\right].$$

For each random variable $\mathrm{Bias4}_e\, F_e L_e$, we have

$$\text{Bias4}_e \, F_e L_e$$

$$= \sum_{t \in \mathcal{T}_e^\ell} \sum_c \Pr(c) \left\langle p_{t,c}, \mathbb{E}\left[ \widetilde{\ell}_{t,c} - \widehat{\ell}_{t,c} \big| \mathcal{H}_{t-1} \right] \right\rangle F_e L_e$$

$$= \sum_{t \in \mathcal{T}_e^\ell} \sum_c \Pr(c) \sum_a p_{t,c}(a) f_e(a) \ell_{t,c}(a) \left( \frac{1}{f_e(a) + \gamma} - \frac{1}{\widehat{f}_e(a) + \frac{3}{2}\gamma} \right) F_e L_e$$

$$= \sum_{t \in \mathcal{T}_e^\ell} \sum_c \Pr(c) \sum_a \widetilde{p}_{t,c}(a) f_e(a) \ell_{t,c}(a) \frac{\widehat{f}_e(a) - f_e(a) + \frac{1}{2}\gamma}{(f_e(a) + \gamma)(\widehat{f}_e(a) + \frac{3}{2}\gamma)} F_e L_e.$$

Thus we have

$$|\text{Bias4}_e \, F_e L_e|$$

$$\leq \left| \sum_{t \in \mathcal{T}_e^\ell} \sum_c \Pr(c) \sum_a \widetilde{p}_{t,c}(a) f_e(a) \ell_{t,c}(a) \frac{\widehat{f}_e(a) - f_e(a) + \frac{1}{2}\gamma}{(f_e(a) + \gamma)(\widehat{f}_e(a) + \frac{3}{2}\gamma)} \right| F_e L_e$$

$$\leq \sum_{t \in \mathcal{T}_e^\ell} \sum_a \widetilde{p}_t(a) \left| \frac{\widehat{f}_e(a) - f_e(a) + \frac{1}{2}\gamma}{\widehat{f}_e(a) + \frac{3}{2}\gamma} \right| F_e L_e$$

$$\leq 8 \sum_{t \in \mathcal{T}_e^\ell} \sum_a \max\left\{ \sqrt{\frac{f_e(a)\iota}{L}}, \frac{\iota}{L} \right\}$$

$$\leq 8L \left( \sqrt{\frac{K\iota}{L}} + \frac{K\iota}{L} \right)$$

$$= 8(\sqrt{KL\iota} + K\iota).$$

Applying Azuma-Hoeffding's inequality to

$$\sum_e \text{Bias4}_e \, F_e L_e - \mathbb{E}\left[ \text{Bias4}_e \, F_e L_e \big| \mathcal{H}_{e-1} \right],$$

we get that for any $\delta > 0$, with probability at least $1 - \delta$, we have

$$\sum_e \text{Bias4}_e \, F_e L_e - \mathbb{E}\left[ \text{Bias4}_e \, F_e L_e \big| \mathcal{H}_{e-1} \right]$$

$$\leq 8(\sqrt{KL\iota} + K\iota)\sqrt{2\frac{T}{L}\log(\frac{\delta}{2})}$$

$$= 8\left( \sqrt{2KT\iota\log(\frac{\delta}{2})} + \sqrt{2\frac{TK^2}{L}\iota\log(\frac{\delta}{2})} \right).$$

Thus we have

$$\sum_e \text{Bias4}_e \, F_e L_e$$

$$\leq 4K\gamma T + \frac{K\exp(-\iota)T}{\gamma} + \frac{98KT\iota}{L} + \frac{\gamma^2 LKT}{\iota}$$

$$+ 8\left( \sqrt{2KT\iota\log(\frac{\delta}{2})} + \sqrt{2\frac{TK^2}{L}\iota\log(\frac{\delta}{2})} \right).$$

with probability at least $\Pr(G) - \delta$.

We then bound the second term

$$\sum_{t \in \mathcal{T}^\ell} \sum_c \Pr(c) \left\langle p_{t,c}, \widetilde{\ell}_{t,c} - \widehat{\ell}_{t,c} - \mathbb{E}\left[\widetilde{\ell}_{t,c} - \widehat{\ell}_{t,c}\middle| \mathcal{H}_{t-1}\right]\right\rangle.$$

For each time step $t \in \mathcal{T}_e^\ell$, we define an indicator function

$$J_t \triangleq \mathbb{1}\left(\forall a, \widetilde{p}_t(a) \le 4 f_e(a)\right).$$

By Lemma 4, event $G$ implies $J_t$.

Similar to previous analysis, we decompose the first term as

$$\sum_{t \in \mathcal{T}^\ell} \sum_c \Pr(c) \left\langle p_{t,c}, \widetilde{\ell}_{t,c} - \widehat{\ell}_{t,c} - \mathbb{E}\left[\widetilde{\ell}_{t,c} - \widehat{\ell}_{t,c}\middle| \mathcal{H}_{t-1}\right]\right\rangle$$
$$= \sum_{t \in \mathcal{T}^\ell} \sum_c \Pr(c) \left\langle p_{t,c}, \widetilde{\ell}_{t,c} - \widehat{\ell}_{t,c} - \mathbb{E}\left[\widetilde{\ell}_{t,c} - \widehat{\ell}_{t,c}\middle| \mathcal{H}_{t-1}\right]\right\rangle F_e J_t$$
$$+ \sum_{t \in \mathcal{T}^\ell} \sum_c \Pr(c) \left\langle p_{t,c}, \widetilde{\ell}_{t,c} - \widehat{\ell}_{t,c} - \mathbb{E}\left[\widetilde{\ell}_{t,c} - \widehat{\ell}_{t,c}\middle| \mathcal{H}_{t-1}\right]\right\rangle (1 - F_e J_t).$$

Since the auxiliary probability vector $\widetilde{p}_{t,c}$ is determined at time $t - 1$, the indicator function $J_t$ is also determined at time $t - 1$. Furthermore, the indicator function $F_e$ is determined at epoch $e - 1$. Thus the product of indicator functions $F_e J_t$ is measurable under filtration $\mathcal{H}_{t-1}$. We have

$$\mathbb{E}\left[\sum_c \Pr(c) \left\langle p_{t,c}, \widetilde{\ell}_{t,c} - \widehat{\ell}_{t,c} - \mathbb{E}\left[\widetilde{\ell}_{t,c} - \widehat{\ell}_{t,c}\middle| \mathcal{H}_{t-1}\right]\right\rangle F_e J_t \middle| \mathcal{H}_{t-1}\right]$$
$$= F_e J_t \mathbb{E}\left[\sum_c \Pr(c) \left\langle p_{t,c}, (\widetilde{\ell}_{t,c} - \widehat{\ell}_{t,c}) - \mathbb{E}\left[(\widetilde{\ell}_{t,c} - \widehat{\ell}_{t,c})\middle| \mathcal{H}_{t-1}\right]\right\rangle \middle| \mathcal{H}_{t-1}\right]$$
$$= 0.$$

Thus the sequence of random variables

$$\left\{\sum_c \Pr(c) \left\langle p_{t,c}, \widetilde{\ell}_{t,c} - \widehat{\ell}_{t,c} - \mathbb{E}\left[\widetilde{\ell}_{t,c} - \widehat{\ell}_{t,c}\middle| \mathcal{H}_{t-1}\right]\right\rangle F_e J_t\right\}_{t \in \mathcal{T}^\ell}$$

forms a martingale difference sequence under the filtration $\{\mathcal{H}_t\}_t$.

We further have that the term

$$\sum_c \Pr(c) \left\langle p_{t,c}, \widetilde{\ell}_{t,c} - \widehat{\ell}_{t,c} - \mathbb{E}\left[\widetilde{\ell}_{t,c} - \widehat{\ell}_{t,c}\middle| \mathcal{H}_{t-1}\right]\right\rangle F_e J_t$$

satisfies

$$
\left| \sum_c \Pr(c) \left\langle p_{t,c}, \widetilde{\ell}_{t,c} - \widehat{\ell}_{t,c} - \mathbb{E}\left[ \widetilde{\ell}_{t,c} - \widehat{\ell}_{t,c} \middle| \mathcal{H}_{t-1} \right] \right\rangle \right| F_e J_t
$$

$$
\leq \left| \sum_c \Pr(c) \sum_a p_{t,c}(a) \ell_{t,c}(a) \left( \frac{\mathbb{1}(a_t = a)}{f_e(a) + \gamma} - \frac{f_e(a)}{f_e(a) + \gamma} \right) \right| F_e J_t
$$

$$
+ \left| \sum_c \Pr(c) \sum_a p_{t,c}(a) \ell_{t,c}(a) \left( \frac{\mathbb{1}(a_t = a)}{\widehat{f}_e(a) + \frac{3}{2}\gamma} - \frac{f_e(a)}{\widehat{f}_e(a) + \frac{3}{2}\gamma} \right) \right| F_e J_t
$$

$$
\leq \sum_c \Pr(c) p_{t,c}(a_t) \frac{\ell_{t,c}(a_t)}{f_e(a_t) + \gamma} F_e J_t
$$

$$
+ \sum_c \Pr(c) \sum_a p_{t,c}(a) \ell_{t,c}(a) \frac{f_e(a)}{f_e(a) + \gamma} F_e J_t
$$

$$
+ \sum_c \Pr(c) p_{t,c}(a_t) \frac{\ell_{t,c}(a_t)}{\widehat{f}_e(a_t) + \frac{3}{2}\gamma} F_e J_t
$$

$$
+ \sum_c \Pr(c) \sum_a p_{t,c}(a) \ell_{t,c}(a) \frac{f_e(a)}{\widehat{f}_e(a) + \frac{3}{2}\gamma} F_e J_t
$$

$$
\leq \left( \frac{p_t(a_t)}{f_e(a_t) + \gamma} + 1 + \frac{p_t(a_t)}{\widehat{f}_e(a_t) + \frac{3}{2}\gamma} + \sum_a p_t(a) \frac{f_e(a)}{\widehat{f}_e(a) + \frac{3}{2}\gamma} \right) F_e J_t
$$

$$
\leq 4 + 1 + 8 + 2 = 15. \tag{Lemma 3}
$$

Applying Azuma-Hoeffding's inequality to the sequnece of random variables

$$
\left\{ \sum_c \Pr(c) \left\langle p_{t,c}, \widetilde{\ell}_{t,c} - \widehat{\ell}_{t,c} - \mathbb{E}\left[ \widetilde{\ell}_{t,c} - \widehat{\ell}_{t,c} \middle| \mathcal{H}_{t-1} \right] \right\rangle F_e J_t \right\}_{t \in \mathcal{T}^\ell},
$$

we get that for any $\delta > 0$, the inequality

$$
\sum_{t \in \mathcal{T}^\ell} \sum_c \Pr(c) \left\langle p_{t,c}, \widetilde{\ell}_{t,c} - \widehat{\ell}_{t,c} - \mathbb{E}\left[ \widetilde{\ell}_{t,c} - \widehat{\ell}_{t,c} \middle| \mathcal{H}_{t-1} \right] \right\rangle F_e J_t \leq 15 \sqrt{T \log(\frac{1}{\delta})}
$$

holds with probability at least $1 - \delta$.

On the other hand, note that event $G$ implies event $F_e J_t$, we get that

$$
\sum_{t \in \mathcal{T}^\ell} \sum_c \Pr(c) \left\langle p_{t,c}, \widetilde{\ell}_{t,c} - \widehat{\ell}_{t,c} - \mathbb{E}\left[ \widetilde{\ell}_{t,c} - \widehat{\ell}_{t,c} \middle| \mathcal{H}_{t-1} \right] \right\rangle (1 - F_e J_t) = 0
$$

whenever event $G$ holds. Thus we get that for any $\delta > 0$, inequality

$$
\sum_{t \in \mathcal{T}^\ell} \sum_c \Pr(c) \left\langle p_{t,c}, \widetilde{\ell}_{t,c} - \widehat{\ell}_{t,c} - \mathbb{E}\left[ \widetilde{\ell}_{t,c} - \widehat{\ell}_{t,c} \middle| \mathcal{H}_{t-1} \right] \right\rangle \leq 15 \sqrt{T \log(\frac{1}{\delta})}
$$

holds with probability at least $\Pr(G) - 2\delta$.

Combining previous results, we get that the following inequality holds with probability at least $\Pr(G) - 2\delta$:

$$\sum_{t \in \mathcal{T}^\ell} \sum_c \Pr(c) \left\langle p_{t,c}, \widetilde{\ell}_{t,c} - \widehat{\ell}_{t,c} \right\rangle$$

$$= \sum_{t \in \mathcal{T}^\ell} \sum_c \Pr(c) \left\langle p_{t,c}, \mathbb{E}\left[ \widetilde{\ell}_{t,c} - \widehat{\ell}_{t,c} \middle| \mathcal{H}_{t-1} \right] \right\rangle$$

$$+ \sum_{t \in \mathcal{T}^\ell} \sum_c \Pr(c) \left\langle p_{t,c}, \widetilde{\ell}_{t,c} - \widehat{\ell}_{t,c} - \mathbb{E}\left[ \widetilde{\ell}_{t,c} - \widehat{\ell}_{t,c} \middle| \mathcal{H}_{t-1} \right] \right\rangle$$

$$\leq 4K\gamma T + \frac{K\exp(-\iota)T}{\gamma} + \frac{98KT\iota}{L} + \frac{\gamma^2 LKT}{\iota}$$

$$+ 8\left( \sqrt{2KT\iota \log(\frac{\delta}{2})} + \sqrt{2\frac{TK^2}{L}\iota \log(\frac{\delta}{2})} \right)$$

$$+ 15\sqrt{T\log(\frac{1}{\delta})}.$$

## B.4. Upper Bounding Remaining Terms

The remaining terms are the $\mathbf{bias}_1$, $\mathbf{bias}_2$, $\mathbf{ftrl}$, and $\mathbf{bias}_3$ terms. These terms are not hard to bound using techniques in standard EXP3-IX analysis (Neu, 2015). We write down these analyses in the sake of completeness.

### B.4.1. UPPER BOUNDING $\mathbf{bias}_1$

Applying Lemma 7 on the loss sequence used in calculating loss estimates, we get that

$$\sum_{t=1}^T \ell_{t,c_t}(a_t) - \ell_{t,c_t}(\pi_{c_t}) - 2 \sum_{t \in \mathcal{T}^\ell} \ell_{t,c_t}(a_t) - \ell_{t,c_t}(\pi_{c_t}) \leq 2\sqrt{T\log(\frac{1}{\delta})}.$$

### B.4.2. UPPER BOUNDING $\mathbf{bias}_2$

Here we bound the $\mathbf{bias}_2$ term

$$\sum_{t \in \mathcal{T}^\ell} \left( \ell_{t,c_t}(a_t) - \ell_{t,c_t}(\pi_{c_t}) - \sum_c \Pr(c)\langle p_{t,c} - \pi_c, \ell_{t,c}\rangle \right).$$

Whenever event $G$ holds, we have $q_t = p_t$. Thus we assume event $G$ holds and replace the $\mathbf{bias}_2$ term by

$$\sum_{t \in \mathcal{T}^\ell} \left( \ell_{t,c_t}(a_t) - \ell_{t,c_t}(\pi_{c_t}) - \sum_c \Pr(c)\langle q_{t,c} - \pi_c, \ell_{t,c}\rangle \right).$$

The new term have the following properties:

- The sequence of random variables

$$\left\{ \ell_{t,c_t}(a_t) - \ell_{t,c_t}(\pi_{c_t}) - \sum_c \Pr(c)\langle q_{t,c} - \pi_c, \ell_{t,c}\rangle \right\}_{t \in \mathcal{T}^\ell}$$

adapts to the filtration $\{\mathcal{H}_t\}_{t \in \mathcal{T}^\ell}$.

- The sequence of random variables satisfies

$$\mathbb{E}\left[ \ell_{t,c_t}(a_t) - \ell_{t,c_t}(\pi_{c_t}) - \sum_c \Pr(c)\langle q_{t,c} - \pi_c, \ell_{t,c}\rangle \middle| \mathcal{H}_{t-1} \right] = 0.$$

- Each random variable satisfies $\ell_{t,c_t}(a_t) - \ell_{t,c_t}(\pi_{c_t}) - \sum_c \Pr(c)\langle q_{t,c} - \pi_c, \ell_{t,c}\rangle \in [-2, 2]$.

By applying Azuma-Hoeffding inequality, we get that for any $\delta \in (0, 1)$, the following inequality holds with probability at least $1 - \delta$:

$$\sum_{t \in \mathcal{T}^\ell} \left( \ell_{t,c_t}(a_t) - \ell_{t,c_t}(\pi_{c_t}) - \sum_c \Pr(c)\langle q_{t,c} - \pi_c, \ell_{t,c}\rangle \right)$$
$$\leq 2\sqrt{T\log(\frac{1}{\delta})}$$

Thus the following inequality holds with probabiity at least $\Pr(G) - \delta$:

$$\sum_{t \in \mathcal{T}^\ell} \left( \ell_{t,c_t}(a_t) - \ell_{t,c_t}(\pi_{c_t}) - \sum_c \Pr(c)\langle p_{t,c} - \pi_c, \ell_{t,c}\rangle \right) \leq 2\sqrt{T\log(\frac{1}{\delta})}.$$

### B.4.3. UPPER BOUNDING **FTRL**

Here we bound the **ftrl** term

$$\sum_{t \in \mathcal{T}^\ell} \sum_c \Pr(c)\langle p_{t,c} - \pi_c, \widehat{\ell}_{t,c}\rangle.$$

By the standard analysis of FTRL algorithms, the **ftrl** term satisfies

$$\sum_{t \in \mathcal{T}^\ell} \sum_c \Pr(c)\langle p_{t,c} - \pi_c, \widehat{\ell}_{t,c}\rangle$$
$$\leq \sum_c \Pr(c) \left( \frac{1}{\eta}\log K + \frac{\eta}{2}\sum_{t \in \mathcal{T}^\ell} \left\langle p_{t,c}, \widehat{\ell}_{t,c}^2 \right\rangle \right).$$

Here $\widehat{\ell}_{t,c}^2$ denotes the vector formed by squaring each component of $\widehat{\ell}_{t,c}$.

By Lemma 3, under event $G$, we have $\widehat{\ell}_{t,c} \leq 2\widetilde{\ell}_{t,c}$. Thus assuming event $G$ holds, we can focus on upper bounding

$$\sum_c \Pr(c) \left( \frac{1}{\eta}\log K + 2\eta\sum_{t \in \mathcal{T}^\ell} \left\langle p_{t,c}, \widetilde{\ell}_{t,c}^2 \right\rangle \right).$$

It suffices to upper bound $\sum_c \Pr(c) \sum_{t \in \mathcal{T}^\ell} \left\langle p_{t,c}, \widetilde{\ell}_{t,c}^2 \right\rangle$. We have

$$\sum_c \Pr(c) \sum_{t \in \mathcal{T}^\ell} \left\langle p_{t,c}, \widetilde{\ell}_{t,c}^2 \right\rangle$$
$$= \sum_c \Pr(c) \sum_{t \in \mathcal{T}^\ell} p_{t,c}(a_t)\widetilde{\ell}_{t,c}^2(a_t)$$
$$= \sum_c \Pr(c) \sum_{t \in \mathcal{T}^\ell} p_{t,c}(a_t)\frac{\ell_{t,c}^2(a_t)}{(f_e(a_t) + \gamma)^2}$$
$$\leq \sum_{t \in \mathcal{T}^\ell} \sum_c \Pr(c)p_{t,c}(a_t)\frac{1}{(f_e(a_t) + \gamma)^2}$$
$$= \sum_{t \in \mathcal{T}^\ell} \frac{p_t(a_t)}{(f_e(a_t) + \gamma)^2}.$$

By Lemma 4, under event $G$, we have

$$\sum_{t \in \mathcal{T}^\ell} \frac{p_t(a_t)}{(f_e(a_t) + \gamma)^2} \leq 2 \sum_{t \in \mathcal{T}^\ell} \frac{1}{f_e(a_t) + \gamma}.$$

We then focus on upper bounding $\sum_{t \in \mathcal{T}^\ell} \frac{1}{f_e(a_t) + \gamma}$.

We have

- The sum of conditional expectations $\sum_{t \in \mathcal{T}^\ell} \mathbb{E}\left[\frac{1}{f_e(a_t) + \gamma} \middle| \mathcal{H}_{t-1}\right] \leq KT$.

- Each term $\frac{1}{f_e(a_t) + \gamma} \leq \frac{1}{\gamma}$.

- The sum of conditional quadratic expectations

$$\sum_{t \in \mathcal{T}^\ell} \mathbb{E}\left[\left(\frac{1}{f_e(a_t) + \gamma}\right)^2 \middle| \mathcal{H}_{t-1}\right]$$

$$= \sum_{t \in \mathcal{T}^\ell} \sum_a \frac{f_e(a)}{(f_e(a) + \gamma)^2}$$

$$\leq \sum_{t \in \mathcal{T}^\ell} \sum_a \frac{1}{f_e(a) + \gamma}.$$

By Freedman's inequality, we have that for any $\lambda \in (0, \gamma]$ and any $\delta \in (0, 1)$, the following inequality holds with probability at least $1 - \delta$:

$$\sum_{t \in \mathcal{T}^\ell} \frac{1}{f_e(a_t) + \gamma} \leq \lambda \sum_{t \in \mathcal{T}^\ell} \sum_a \frac{1}{f_e(a) + \gamma} + \frac{1}{\lambda} \log\left(\frac{1}{\delta}\right) + KT.$$

We pick $\lambda = \gamma$ to get that

$$\sum_{t \in \mathcal{T}^\ell} \frac{1}{f_e(a_t) + \gamma} \leq 2KT + \frac{1}{\gamma} \log\left(\frac{1}{\delta}\right)$$

with probability at least $1 - \delta$.

Substituting this inequality back, we get that the folowing inequality holds with probability at least $\Pr(G) - \delta$:

$$\sum_c \Pr(c) \left(\frac{1}{\eta} \log K + \frac{\eta}{2} \sum_{t \in \mathcal{T}^\ell} \left\langle p_{t,c}, \widehat{\ell}_{t,c}^2 \right\rangle\right)$$

$$\leq \frac{1}{\eta} \log K + 4\eta \sum_{t \in \mathcal{T}^\ell} \frac{1}{f_e(a_t) + \gamma}$$

$$\leq \frac{1}{\eta} \log K + 4\eta \left(2KT + \frac{1}{\gamma} \log\left(\frac{1}{\delta}\right)\right).$$

### B.4.4. UPPER BOUNDING $\mathbf{bias}_3$

Here we bound the $\mathbf{bias}_3$ term

$$\sum_{t \in \mathcal{T}^\ell} \sum_c \Pr(c) \left\langle p_{t,c}, \ell_{t,c} - \widetilde{\ell}_{t,c} \right\rangle.$$

We decompose it as

$$\sum_{t \in \mathcal{T}^\ell} \sum_c \Pr(c) \left\langle p_{t,c}, \ell_{t,c} - \widetilde{\ell}_{t,c} \right\rangle$$

$$= \sum_{t \in \mathcal{T}^\ell} \sum_c \Pr(c) \left\langle p_{t,c}, \ell_{t,c} - \mathbb{E}\left[\widetilde{\ell}_{t,c} \middle| \mathcal{H}_{t-1}\right] \right\rangle$$

$$+ \sum_{t \in \mathcal{T}^\ell} \sum_c \Pr(c) \left\langle p_{t,c}, \mathbb{E}\left[\widetilde{\ell}_{t,c} \middle| \mathcal{H}_{t-1}\right] - \widetilde{\ell}_{t,c} \right\rangle.$$

The first component satisfies

$$
\sum_{t \in \mathcal{T}^\ell} \sum_c \Pr(c) \left\langle p_{t,c}, \ell_{t,c} - \mathbb{E}\left[\widetilde{\ell}_{t,c} \middle| \mathcal{H}_{t-1}\right] \right\rangle
$$

$$
= \sum_{t \in \mathcal{T}^\ell} \sum_c \Pr(c) \sum_a p_{t,c}(a)\ell_{t,c}(a) \frac{\gamma}{f_e(a) + \gamma}
$$

$$
\leq \sum_{t \in \mathcal{T}^\ell} \sum_a p_t(a) \frac{\gamma}{f_e(a) + \gamma}.
$$

Assuming event $G$ holds, we have

$$
\sum_{t \in \mathcal{T}^\ell} \sum_a p_t(a) \frac{\gamma}{f_e(a) + \gamma}
$$

$$
\leq 2 \sum_{t \in \mathcal{T}^\ell} \sum_a \gamma \leq 2KT\gamma. \tag{Lemma 4}
$$

We then bound the second component $\sum_{t \in \mathcal{T}^\ell} \sum_c \Pr(c) \left\langle p_{t,c}, \mathbb{E}\left[\widetilde{\ell}_{t,c} \middle| \mathcal{H}_{t-1}\right] - \widetilde{\ell}_{t,c} \right\rangle$. For each time step $t \in \mathcal{T}^\ell$, we define an indicator function

$$
L_t \triangleq \mathbb{1}\left(\forall a, p_t(a) \leq 4f_e(a)\right).
$$

By Lemma 4, event $G$ implies event $L_t$. Thus we have

$$
\sum_{t \in \mathcal{T}^\ell} \sum_c \Pr(c) \left\langle p_{t,c}, \mathbb{E}\left[\widetilde{\ell}_{t,c} \middle| \mathcal{H}_{t-1}\right] - \widetilde{\ell}_{t,c} \right\rangle
$$

$$
= \sum_{t \in \mathcal{T}^\ell} \sum_c \Pr(c) \left\langle p_{t,c}, \mathbb{E}\left[\widetilde{\ell}_{t,c} \middle| \mathcal{H}_{t-1}\right] - \widetilde{\ell}_{t,c} \right\rangle L_t.
$$

under event $G$. We then assume event $G$ holds and focus on upper bounding

$$
\sum_{t \in \mathcal{T}^\ell} \sum_c \Pr(c) \left\langle p_{t,c}, \mathbb{E}\left[\widetilde{\ell}_{t,c} \middle| \mathcal{H}_{t-1}\right] - \widetilde{\ell}_{t,c} \right\rangle L_t.
$$

Since the probability vector $p_{t,c}$ is determined at time $t - 1$, the indicator function $L_t$ is also determined at time $t - 1$. Thus the summand random variable

$$
\sum_c \Pr(c) \left\langle p_{t,c}, \mathbb{E}\left[\widetilde{\ell}_{t,c} \middle| \mathcal{H}_{t-1}\right] - \widetilde{\ell}_{t,c} \right\rangle L_t
$$

satisfies

$$
\mathbb{E}\left[\sum_c \Pr(c) \left\langle p_{t,c}, \mathbb{E}\left[\widetilde{\ell}_{t,c} \middle| \mathcal{H}_{t-1}\right] - \widetilde{\ell}_{t,c} \right\rangle L_t \middle| \mathcal{H}_{t-1}\right]
$$

$$
= \mathbb{E}\left[\sum_c \Pr(c) \left\langle p_{t,c}, \mathbb{E}\left[\widetilde{\ell}_{t,c} \middle| \mathcal{H}_{t-1}\right] - \widetilde{\ell}_{t,c} \right\rangle \middle| \mathcal{H}_{t-1}\right] L_t
$$

$$
= 0.
$$

Thus the sequence of random varialbes

$$
\left\{\sum_c \Pr(c) \left\langle p_{t,c}, \mathbb{E}\left[\widetilde{\ell}_{t,c} \middle| \mathcal{H}_{t-1}\right] - \widetilde{\ell}_{t,c} \right\rangle L_t\right\}_{t \in \mathcal{T}^\ell}
$$

forms a martingale difference sequence with respect to the filtration $\{\mathcal{H}_t\}_{t \in \mathcal{T}^\ell}$.

The summand random variable further satisfies

$$\sum_c \Pr(c) \left\langle p_{t,c}, \widetilde{\ell}_{t,c} \right\rangle L_t$$

$$= \sum_c \Pr(c) p_{t,c}(a_t) \frac{\ell_{t,c}(a_t)}{f_e(a_t) + \gamma} L_t$$

$$\leq \sum_c \Pr(c) p_{t,c}(a_t) \frac{1}{f_e(a_t) + \gamma} L_t$$

$$= \frac{p_t(a_t)}{f_e(a_t) + \gamma} L_t$$

$$\leq 4.$$

Then by Azuma-Hoeffding inequality, the following inequality holds with probability at least $1 - \delta$:

$$\sum_{t \in \mathcal{T}^\ell} \sum_c \Pr(c) \left\langle p_{t,c}, \mathbb{E}\left[\widetilde{\ell}_{t,c} \,\middle|\, \mathcal{H}_{t-1}\right] - \widetilde{\ell}_{t,c} \right\rangle L_t \leq 4\sqrt{T \log(\frac{1}{\delta})}.$$

Thus we have

$$\sum_{t \in \mathcal{T}^\ell} \sum_c \Pr(c) \left\langle p_{t,c}, \mathbb{E}\left[\widetilde{\ell}_{t,c} \,\middle|\, \mathcal{H}_{t-1}\right] - \widetilde{\ell}_{t,c} \right\rangle \leq 4\sqrt{T \log(\frac{1}{\delta})}$$

with probability at least $\Pr(G) - \delta$.

Combining the first component and the second component, we get that

$$\sum_{t \in \mathcal{T}^\ell} \sum_c \Pr(c) \left\langle p_{t,c}, \ell_{t,c} - \widetilde{\ell}_{t,c} \right\rangle \leq 4\sqrt{T \log(\frac{1}{\delta})} + 2KT\gamma$$

with probability at least $\Pr(G) - \delta$.

### B.5. Combining the pieces

Combining all previous arguments, we get that

$$\mathrm{Reg}(\pi)$$

$$\leq 2\sqrt{T \log(\frac{1}{\delta})}$$

$$+ 4\sqrt{T \log(\frac{1}{\delta})}$$

$$+ 2\frac{1}{\eta} \log K + 8\eta \left(2KT + \frac{1}{\gamma} \log(\frac{1}{\delta})\right)$$

$$+ 8\sqrt{T \log(\frac{1}{\delta})} + 4KT\gamma$$

$$+ 8K\gamma T + 2\frac{K \exp(-\iota)T}{\gamma} + 2\frac{98KT\iota}{L} + 2\frac{\gamma^2 LKT}{\iota}$$

$$+ 16 \left(\sqrt{2KT\iota \log(\frac{\delta}{2})} + 2\sqrt{2\frac{TK^2}{L}\iota \log(\frac{\delta}{2})}\right)$$

$$+ 30\sqrt{T \log(\frac{1}{\delta})}$$

$$+ 2(\frac{8 \cdot 36\iota}{\gamma} + 2\frac{8}{\gamma}) \log(1/\delta) + 2KT \exp(-\iota)$$

with probability at least $\Pr(G) - 8\delta$.

Taking $\iota = 2\log(8KT\frac{1}{\delta})$, $L = \sqrt{\frac{\iota KT}{\log(K)}} = \widetilde{\Theta}(\sqrt{KT\log\frac{1}{\delta}})$, $\gamma = \frac{16\iota}{L} = \widetilde{\Theta}(\sqrt{\frac{\log(1/\delta)}{KT}})$, and $\eta = \frac{\gamma}{2(2L\gamma+\iota)} = \widetilde{\Theta}(1/\sqrt{KT\log(1/\delta)})$, it is easy to see that $\mathrm{Reg}(\pi) = \widetilde{O}(\sqrt{KT\log\frac{1}{\delta}})$ with probability at least $\Pr(G) - 8\delta \geq 1 - 9\delta$ for any policy $\pi$ and any $\delta \in (0,1)$.

The final step is rescaling the probability constant by a factor of $1/9$, which gives that $\mathrm{Reg}(\pi) = \widetilde{O}(\sqrt{KT\log\frac{1}{\delta}})$ with probability at least $1 - \delta$ and ends the proof of Theorem 1.

