# OpenReview forum: "High Probability Bound for Cross-Learning Contextual Bandits with Unknown Context Distributions"
_ICML.cc/2025/Conference — ICML 2025 poster_

### Official Review · Reviewer_McaC · 2025-03-09

**Overall Recommendation:** 3

**Summary:**

The paper studies the setting of cross-learning contextual bandit problem and provides a new analysis to the algorithm in Schneider & Zimmert (2023), showing the algorithm can achieve near optimal bound in high probability. They utilizes the weak dependency structure between different epochs and this approach might have chance to generalize to other analysis of algorithm.

## update after rebuttal
I thank the authors for the response. Nothing major has changed, and I keep my score.

**Claims And Evidence:**

Yes

**Essential References Not Discussed:**

Nothing particular.

**Experimental Designs Or Analyses:**

They don’t have an experiment part.

**Methods And Evaluation Criteria:**

Yes

**Other Comments Or Suggestions:**

The line 319 seems overfull.

**Other Strengths And Weaknesses:**

They provide a new analysis and improve the previous result, from expectation bound to a high probability bound.

**Questions For Authors:**

The paper gives us a new analysis of algorithm in Schneider & Zimmert (2023) and I wonder whether this analysis approach can be applied to other bandit algorithms.

**Relation To Broader Scientific Literature:**

The paper provided theoretical improvement for an algorithm. If the technique can generalize to other algorithms, then the paper can be considered as a good contribution.

**Theoretical Claims:**

Yes, I check the main theorem and its proof.

---

> ### Author Rebuttal · Authors · 2025-03-29
>
> Dear review McaC:
>
> Thank you for your positive feedback. We answer your question below.
>
> ---
>
> **Question: The line 319 seems overfull**
>
> Thanks for pointing this out. We will fix it in the new version.
>
> ---
>
> **Question:  whether this analysis approach can be applied to other bandit algorithms**
>
> This is an good question. We believe that this analysis could potentially be applied to other algorithms that have an epoch-based execution structure and have weakly dependent structures between epochs. We will further investigate this question in future research.
>
> ---
>
> Once again, we sincerely thank you for your positive review. We hope our response addresses your concern.

---

### Official Review · Reviewer_m6dx · 2025-03-11

**Overall Recommendation:** 3

**Summary:**

This paper considers contextual bandits with cross learning, where the learner observes the loss associated with the action across all possible contexts. The losses are adversarial, but the contexts are stochastic.
The authors improve the analysis of Schneider & Zimmert (2023) to prove high probability regret bound, rather than the known expected regret bound, to the same algorithm. They do that by making use of the weak dependency structure between different epochs, which was overlooked innprevious analyses. Additionally, they refine martingale inequalities to complete their analysis.

**Claims And Evidence:**

Yes, they prove high probability regret gurantee.

**Essential References Not Discussed:**

No.

**Experimental Designs Or Analyses:**

I read the analysis sketch appears in the main paper, seems OK.

**Methods And Evaluation Criteria:**

The authors consider context regret, which is appropriate measure of perfirmance.

**Other Comments Or Suggestions:**

Line 319 left – overflow in inline equation.

**Other Strengths And Weaknesses:**

Strengths:

1.	The authors present novel and improved analysis to a known algorithm.
2.	Results seems sound.
3.	Well-written paper.

Weakness:

My worries are about the novelty – which is only the analysis; and about the broader impact of this work – are those techniques will be useful in other settings?

**Questions For Authors:**

See weakness.

**Relation To Broader Scientific Literature:**

Not sure that exists relation to broader literature.

**Theoretical Claims:**

I read the analysis sketch appears in the main paper, seems OK.

---

> ### Author Rebuttal · Authors · 2025-03-29
>
> Dear Reviewer m6dx,
>
> Thank you for your positive feedback. We answer your question below.
>
> ---
>
> **Question: the novelty and the broader impact**
>
> The reviewer questioned the novelty of our work, particularly noting that our contribution appears to be “only the analysis.” We would like to clarify this point. First, in the field of adversarial bandit research, providing a high-probability bound alone is often regarded as a substantial contribution. Second, we respectfully argue that enhancing the results of an existing algorithm through novel analysis—achieving the same outcome with a simpler approach—can, in a certain sense, be considered an advantage over proposing a more complex algorithm. Finally, our work demonstrates technical novelty, as decomposing regret by epochs is a relatively uncommon approach in high-probability bound studies.
>
> Regarding the broader impact of our work, the reviewer asked, “Are those techniques useful in other settings?” This is an excellent question and aligns with the direction we aim to explore next. We believe our techniques could be beneficial to other algorithms that involve epoch-based subroutine and exhibit weakly dependent relationships between epochs. We are committed to further investigating this question.
>
> ---
>
> Once again, we sincerely thank you for your positive review. We hope our response addresses your concern.

---

### Official Review · Reviewer_JF8C · 2025-03-12

**Overall Recommendation:** 3

**Summary:**

This paper provides a high probability bound for the cross-learning problem in Schneider & Zimmert, 2023, where a regret bound is provided in expectation.

**Claims And Evidence:**

N/A

**Essential References Not Discussed:**

N/A

**Experimental Designs Or Analyses:**

N/A

**Methods And Evaluation Criteria:**

N/A

**Other Comments Or Suggestions:**

N/A

**Other Strengths And Weaknesses:**

Strengths:
1. I appreciate the detailed description of the literature and I think the current presentation is almost clear (but see my comments below).

Weakness:
1. On line 275, the authors mentioned that a term of $\sum\sum Pr(c) \langle \pi_c, \tilde{l}_{t, c} - l_{t, c}\rangle$ is saved. However, the significance of this term is not discussed. If the same technique in Schneider & Zimmert, 2023 is used to derive the high probability regret bound, will this term be of non-trivial order that will lead to a term worse than sqrt(KT) and eventually lead to a nonoptimal bound? Or is this term hard to bound given the technique in Schneider & Zimmert, 2023? Or is there any other terms that cause difficulty for the technique in the prior literature? Some discussions would be helpful for me to understand.

**Questions For Authors:**

N/A

**Relation To Broader Scientific Literature:**

N/A

**Theoretical Claims:**

N/A

---

> ### Author Rebuttal · Authors · 2025-03-29
>
> Dear Reviewer JF8C,
>
> Thank you for your positive feedback. We answer your question below.
>
> ---
>
> **Question: the significance of this term $\sum \sum Pr(c)  <\pi_c, \tilde{l} _{t, c} -  l_{t, c} >$**
>
> To achieve a high-probability bound, saving this term is essential. In Schneider & Zimmert (2023), this term is not saved. Thus they need to bound the term $\sum\sum Pr(c) < \pi_c, \hat{l}_{t, c} - \tilde{l}_{t, c} >$ (implicitly in their $\text{bias}_2$ term). As a result, their approach yields a bound in expectation only. In our work, saving the term $\sum \sum Pr(c) < \pi_c, \tilde{l}_{t, c} - l_{t, c} >$ allows us to counteract the random deviation, thereby strengthening the expectation bound into a high-probability bound.
>
> ---
>
> Once again, we sincerely thank you for your positive review. We hope our response addresses your concern.

---

### Official Review · Reviewer_Kzo3 · 2025-03-14

**Overall Recommendation:** 3

**Summary:**

The submission studies contextual bandits with cross-learning, where the feedback information is the loss of the chosen action in all contexts. The main result is a regret bound that holds in high probability, which improves on a previous bound that holds only in expectation (Schneider & Zimmert 2023).


## Update after rebuttal

The authors' reply addressed my concerns. In light of this, I have revised my recommendation to weak accept.

**Claims And Evidence:**

Yes.

**Essential References Not Discussed:**

I am fine with the related work discussed in Section 1.2.

**Experimental Designs Or Analyses:**

There is no experimental result; the submission is a theoretical paper.

**Methods And Evaluation Criteria:**

The technical contributions are as follows.
 - The submission proposed a novel way to decompose the regret. The new term contributes a negative term (lines 297R and 637) to make the high probability bound possible.
 - The newly introduced term is unbounded, hence, a surrogate indicator function (lines 324R and 452) is designed to make the sequence of epoch values a martingale difference sequence.

With the two modifications above, Freedman’s inequality can be applied to provide a high probability bound.

**Other Comments Or Suggestions:**

Please see the comments in the above parts.

**Other Strengths And Weaknesses:**

Weaknesses
- Although clever, the impact of the techniques developed is unclear. Currently, it seems to be a specific treatment to a specific algorithm and seems to be incremental (from the perspective of the result). That is, the impact of the proposed technique on the theoretical community is unclear.
- The main context mentioned several practical cross-learning applications. It is not clear whether the algorithm developed in this line of research (Schneider & Zimmert 2023) outperforms conventional contextual bandit algorithms. An empirical verification on realistic datasets would reinforce the need to develop a deeper theoretical understanding.
- Cross-learning is not defined mathematically in section 2 [110R].

**Questions For Authors:**

What is the mathematical meaning of "weakly dependent" [110R]? Where is it used in the derivation (which inequality in the appendix)?

**Relation To Broader Scientific Literature:**

The way the unbounded sequence is handled, which makes martingale concentration analysis possible, could benefit concentration analysis.

**Theoretical Claims:**

I have read appendices A, B1, and B2 and browsed through the rest of Appendix B.

---

> ### Author Rebuttal · Authors · 2025-03-28
>
> Dear Reviewer Kzo3,
>
> We sincerely appreciate your valuable suggestions and thoughtful feedback. Below, we address each of your concerns in detail.
>
> ---
>
> **Question: The impact of our results on the theoretical community**
>
> The reviewer noted that “the result seems to be a specific treatment to a specific algorithm and seems to be incremental.” We respectfully disagree with this assessment. We would like to emphasize that in the field of adversarial bandit research, providing high-probability bounds is a widely recognized and significant contribution in its own right (e.g., [1, 2, 3]). From a technical perspective, our argument for the high-probability bound is novel. Specifically, we decompose the regret across different epochs and leverage the weakly dependent structures between these epochs—a method that, to the best of our knowledge, is not commonly employed in existing high-probability bounds for adversarial bandits.
>
> ---
>
> **Question:  the suggestion for empirical verification on realistic datasets**
>
> We thank the reviewer for raising this point, as it highlights an opportunity to strengthen our work. In response, we will include experimental data in the revised version of the paper to reinforce the need for a deeper theoretical understanding and to complement our theoretical findings.
>
> ---
>
> **Question: the mathematical definition of cross-learning in Section 2 [110R]**
>
> We are grateful to the reviewer for pointing out this oversight. While we provided an explanation of cross-learning in [122R], we acknowledge that a clearer presentation is warranted. Based on your suggestion, we will introduce a standalone definition of cross-learning in the revised version to ensure this concept is explicitly and effectively conveyed.
>
> ---
>
> **Question: What is the mathematical meaning of "weakly dependent" [110R]? Where is it used in the derivation (which inequality in the appendix)?**
>
> "Weakly dependent" is an informal term commonly used in the context of concentration inequalities([4]). It generally refers to a sequence of random variables $X_i$ that, while not fully independent, forms a martingale, allowing us to derive concentration inequalities akin to those for independent random variables. In our work, it means that we can decompose the regret into epochs and  the regrets of different epochs form an martingale sequence.
>
> This property is utilized wherever epoch-based decomposition appears in our proofs. For example, it is applied in bounding $\sum_e \text{Bias5}_e$ in Appendix B.2 and $\sum_e \text{Bias4}_e$ in Appendix B.3.
>
> ---
>
> Once again, we express our gratitude to the reviewer for their insightful and valuable feedback. These comments have greatly assisted us in refining the presentation of our paper and better articulating its contributions. We hope that our responses adequately address your concerns and that you will consider increasing your support for our work in light of these revisions.
>
> ---
>
> References:
>
> [1] Luo, H., Tong, H., Zhang, M., & Zhang, Y. (2022). Improved High-Probability Regret for Adversarial Bandits with Time-Varying Feedback Graphs. International Conference on Algorithmic Learning Theory.
>
> [2] Neu, G. (2015). Explore no more: Improved high-probability regret bounds for non-stochastic bandits. Neural Information Processing Systems.
>
> [3] Bartlett, P.L., Dani, V., Hayes, T.P., Kakade, S.M., Rakhlin, A., & Tewari, A. (2008). High-Probability Regret Bounds for Bandit Online Linear Optimization. Annual Conference Computational Learning Theory.
>
> [4] Pelekis, C., & Ramon, J. (2015). Hoeffding's inequality for sums of weakly dependent random variables. arXiv: Probability.

---

> > ### Comment · Reviewer_Kzo3 · 2025-04-02
> >
> > Question: The impact of our results on the theoretical community
> >
> > Thank you for your reply. I wasn't clear enough. Of course it is a contribution to provide a bound that holds with high probability. My point was, can your method be extended to analyse other algorithms/problems, or does your method simply provide "yet another" high probability bound?
> >
> >
> > Question: What is the mathematical meaning of "weakly dependent" [110R]?
> >
> > Thank you very much. I think this answer is very helpful in understanding the logic behind your analysis. Please include it in the paper.

---

> > > ### Author Response · Authors · 2025-04-02
> > >
> > > Dear reviewer Kzo3,
> > >
> > > Thanks for your timely reply.
> > >
> > > About the impact of our results on the theoretical community. Currently we have not identified a direct additional application of our method beyond what is currently presented. However, we believe that the techniques developed in our work have the potential to be applied to other algorithms with multiple epochs and a weakly dependent structure between epochs. The reason we believe this comes from the simplicity and generality of the key structure underlying our analysis—namely, the weakly dependent structure between epochs—which we believe may appear in other algorithms. We view this as a promising direction for future investigation and plan to explore it further in subsequent studies.
> > >
> > > About the mathematical meaning of "weakly dependent". Thanks for your recognition, we will include a new paragraph explaining this term in the new version of our paper.
> > >
> > > Thank you once again for your valuable input. We hope that our responses adequately address your concerns and that you will consider increasing your support for our work in light of these revisions.

---

### Decision · Program_Chairs · 2025-05-01

**Decision:**

Accept (poster)

**Comment:**

This paper gives a refined analysis for a recent algorithm on contextual bandits with cross-context feedback. This kind of problem subsumes for example sleeping bandits.

Using a novel martingale analysis to deal with weakly dependent traces over the epochs in the algorithm allows them to show high probability bounds instead of in-expectation bounds.

Overall, all reviewers agree that this is a nice contribution and that the proofs are sound. None of them is overly excited and gives a strong vote for acceptance. The application is somewhat niche and it is unclear how much interest there is from the broader community.